# THE QUIET PROMPT: ERASING INEFFABLE STYLES FROM DIFFUSION VIA CONCEPT EMBEDDING

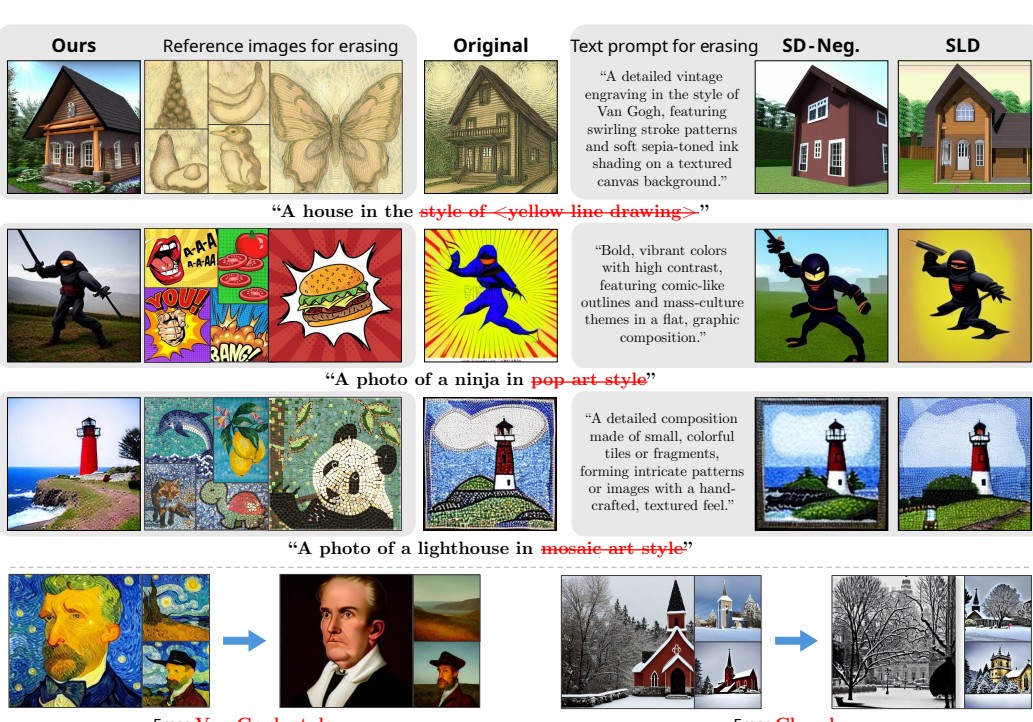

Figure 1: Qualitative comparison between our reference image-based concept erasure (left) and text-based content removal methods (right). In the "Ours-Reference images for erasing" column, (fewer than ten) reference images are used to suppress complex, ineffable (i.e., difficult-to-describe) styles (top three rows) or object concepts (bottom right). The "Original" column shows unaltered outputs (e.g., "A house in the style of <yellow line drawing>"), and "Text prompt for erasing" column presents verbose text-based removal. Unlike SD-Neg. Ho & Salimans (2022) and SLD Schramowski et al. (2023), which leave style residues or artifacts, our method clearly suppresses undesired styles and objects while preserving image quality and text-image alignment. The bottom row shows our method only: on the left, the "A portrait of a character in a scenic environment by Vincent van Gogh style" image and its Van Gogh–style removal; on the right, the "Church with snowy background" image and its church-object removal.

## ABSTRACT

Recent text-to-image diffusion models (DMs) depend on vast, uncurated datasets that often include copyrighted or personal images, risking the generation of unwanted content. Fully curating these datasets is costly, and retraining the entire DMs from scratch is impractical. To address this, concept erasure methods have emerged to suppress undesirable outputs in pre-trained models without additional training. However, because these methods solely rely on text prompts for erasure, they are largely limited to concepts that can be explicitly described in words. This raises an important question: how can we erase highly personalized or visually

nuanced concepts that are difficult to articulate verbally? To tackle this issue, we leverage fewer than ten reference images to derive a unified concept embedding that seamlessly captures even ineffable styles or content. During inference, the concept embedding serves as negative guidance—facilitating both the precise removal of complex visual moods and comprehensive erasure of target concepts without verbose prompts. Our method consists of two stages: (1) **Concept embedding generation**, which compresses the color, texture, and morphology of an unwanted style into a single latent vector; and (2) **Concept-aware negative guidance**, which uses the extracted concept embedding as a semantic anchor to steer the diffusion process away from the undesirable concept without requiring model retraining. Building on these components, we propose Quiet Prompt (QuP), a method that simultaneously delivers consistent style removal—effectively erasing complex styles without verbose text—and comprehensive concept removal, allowing broad specification of target concepts without enumerating multiple synonyms or descriptions. Extensive evaluations on idiosyncratic and historical art styles, as well as object removal tasks, demonstrate that our method achieves superior performance in both unwanted style suppression—measured by a novel style score—and content preservation, outperforming state-of-the-art text-based baselines.

# 1 INTRODUCTION

Recently, diffusion models (DMs) have gained significant attention for their ability to generate photorealistic images. As a result, the task of text-to-image (T2I) generation—where text serves as a conditioning signal—has seen rapid advances Baumhauer et al. (2022); Ding et al. (2022); Saharia et al. (2022); Rombach et al. (2022); Ramesh et al. (2021); Wu et al. (2022); Nichol et al. (2021); Yu et al. (2022); Betker et al. (2023). The progress of large-scale T2I models relies on massive training datasets crawled from various platforms, most of which are uncurated Panda & Prathosh (2024); Evans et al. (2024); Gadre et al. (2023); Zhang et al. (2024d); Bianchi et al. (2023); Schuhmann et al. (2022); Smith et al. (2023); Luccioni et al. (2023). However, such uncurated training datasets may include personally identifiable information, explicit content, and copyrighted artworks Garcia et al. (2023); Bender et al. (2021); Somepalli et al. (2023a); Qu et al. (2023); Andrews et al. (2023); Birhane et al. (2023). Consequently, DMs trained on these datasets are at risk of indiscriminately mimicking the personalized styles of individual artists or generating not safe for work (NSFW) content Sohn et al. (2023); Somepalli et al. (2023b); Ruiz et al. (2023). Every individual has the right to prevent their creative works from being used in the training of generative models. However, retraining large-scale DMs from scratch for each user request is computationally infeasible. This has led to the emergence of the concept erasure task, which aims to suppress undesired content generation in pre-trained DMs without requiring additional training Gandikota et al. (2023; 2024); Lu et al. (2024); Gong et al. (2024).

Existing style removal methods Schramowski et al. (2023); Zhang et al. (2024b); Kumari et al. (2023); Gandikota et al. (2023); Lyu et al. (2024); Gandikota et al. (2024); Zhang et al. (2024c) without additional training typically block the generation of undesirable content by specifying the target concept through text prompts as shown in the right panel of the top three rows in Fig. 1. However, such text-based approaches often require domain knowledge to describe the style to be removed accurately, and struggle to capture subtle style elements–such as color palettes or lighting conditions–using text alone. Furthermore, as prompts become longer in order to describe more complex styles, minor variations in wording, word order, or emphasis can significantly affect the resulting embeddings. As a result, even semantically equivalent prompts may yield inconsistent embeddings, making it difficult to consistently suppress the intended style.

To address these limitations, as shown in the left panel of the top three rows in Fig. 1, we propose a novel training-free style removal method that effectively identifies complex styles using only a small number of reference images. Specifically, we estimate a concept embedding that intuitively represents the concept or style to be removed, based on a few reference images. During inference, this concept embedding is used as negative guidance. That is, the concept embedding derived from the reference images is supplied as a negative prompt to the DM, guiding the DM to interpret features associated with the embedding as undesirable. In other words, the semantics of the concept embedding are treated as negative conditions during generation, thereby suppressing the appearance of the target concept or

style in the output. This allows us to accurately and effectively suppress rich stylistic attributes–such as color tone, texture, and morphological features–without relying on prompt engineering. Moreover, while prior methods must manually enumerate synonyms such as "car" and "vehicle" to remove the concept of "automobile", our approach automatically captures and suppresses the relevant concept using only a few example images. The main contributions of our method are as follows:

**(1) Indescribable style removal:** We enable the removal of styles that are difficult to articulate with language by identifying them from a few reference images, allowing complex visual moods and forms to be suppressed without verbose prompts.

**(2) Comprehensive concept removal:** Our method learns a concept embedding from a small number of reference images, enabling the suppression of a target concept without explicitly specifying all its synonymous or paraphrased forms.

**(3) Novel style removal evaluation metric:** We introduce a new, intuitive metric for the quantitative evaluation of style removal, filling a gap left by the lack of standardized evaluation criteria in prior works.

## 2 RELATED WORKS

Generative models trained on large-scale uncurated datasets crawled from various online platforms often reproduce personalized images or imitate artistic styles present in the training data. In this section, we provide a broad overview of prior research across two related areas aimed at addressing these challenges: unlearning methods, which aim to make models forget specific data; and concept removal methods, which suppress the generation of inappropriate contents (e.g., violence and nudity).

**Memorization & unlearning.** Generative models (GMs) often exhibit "regurgitation", where training samples are reproduced verbatim during generation van den Burg & Williams (2021); Jalali et al. (2023); Theis et al. (2015); Zhao et al. (2018). This raises serious privacy and copyright concerns, prompting the development of unlearning methods to remove sensitive data Bourtoule et al. (2021). Among inference-time approaches, Baumhauer et al. Baumhauer et al. (2022) apply a linear projection to eliminate class directions, while Panda et al. Panda & Prathosh (2024) filter outputs by projection similarity with negative samples. Kumari et al. Kumari et al. (2023) propose exact unlearning by directly modifying model parameters. However, these methods focus on specific data points or classes, falling short in suppressing higher-level stylistic attributes. Thus, concept erasure methods capable of controlling such attributes are essential Kumari et al. (2023); Bau et al. (2020).

**Concept removal.** This section addresses methods for preventing the generation of broader concepts such as styles or offensive imagery. Safe Latent Diffusion (SLD) Schramowski et al. (2023) alters classifier-free guidance during inference to suppress explicit content. Erasing Diffusion Models (ESD) Gandikota et al. (2023) identifies unwanted artistic styles through text descriptions and modifies model weights to suppress the components that are activated in response to the target style. Ablating Concepts (AC) Kumari et al. (2023) replaces a target concept with an anchor concept to redirect generation. Mass Concept Erasure (MACE) Lu et al. (2024) combines closed-form cross-attention refinement, which remaps attention projections from the target concept to the anchor concept, with LoRA modules Hu et al. (2022). Unlike these approaches, which often require many image-text pairs or focus on simple styles Zhang et al. (2024a); Hao et al. (2023); Sridhar et al. (2024); Zhou et al. (2022); Liu & Chilton (2022), our work targets suppression of complex styles involving high-level visual properties.

## 3 METHOD

### 3.1 PRELIMINARIES

**Latent diffusion models** (LDMs) Rombach et al. (2022) extend diffusion models (DMs) Ho et al. (2020) by enhancing efficiency and perceptual quality. Specifically, stable diffusion (SD) conducts the diffusion process within a lower-dimensional latent space $\mathbf{z}$ derived from a variational autoencoder (VAE) $\mathcal{E}$ Kingma et al. (2013). Let $\tau_\theta(\mathbf{c}_p)$ be a text encoding that maps a text condition $\mathbf{c}_p$ into

conditioning embeddings. The loss between added noise ($\epsilon \sim \mathcal{N}(0, \mathbf{I})$) and predicted noise ($\epsilon_\theta(\cdot)$) of LDM is defined as follows:

$$\mathcal{L}_{\text{LDM}} := \mathbb{E}_{\mathcal{E}(x), \mathbf{c}_p, \epsilon \sim \mathcal{N}(0, \mathbf{I}), t}\big\| \epsilon - \epsilon_\theta\big(\mathbf{z}_t, t, \tau_\theta(\mathbf{c}_p)\big)\big\|_2^2, \tag{1}$$

where $\mathbf{z}_t$ is the noised version of latent at timestep $t$ and $\epsilon_\theta$ denotes the denoising U-Net Ronneberger et al. (2015) trained to reconstruct the original latent image. The initial latent vector $z_0 = \mathcal{E}(x)$ is obtained by encoding the input image $x$ using the VAE encoder $\mathcal{E}$.

In LDM, classifier-free guidance (CFG) Ho & Salimans (2022) is adopted as a conditioning technique that guides image generation to match the text prompt without relying on an externally trained classifier. This method combines an unconditioned $\epsilon_\theta$-prediction and a $\epsilon_\theta$-conditioned prediction during inference, effectively steering the DM's Ho et al. (2020) predicted distribution from unconditional to conditional. The guided noise prediction ($\tilde{\epsilon}_\theta$) is defined as:

$$\tilde{\epsilon}_\theta(\mathbf{z}_t, \mathbf{c}_p) := \epsilon_\theta(\mathbf{z}_t) + s_g\big(\epsilon_\theta(\mathbf{z}_t, \tau_\theta(\mathbf{c}_p)) - \epsilon_\theta(\mathbf{z}_t)\big), \tag{2}$$

where guidance scale $s_g$ is typically chosen as $s_g \in (0, 20]$ Schramowski et al. (2023). This allows the DM to generate high-quality images that remain faithful to the text condition. CFG interprets the meaning of a text prompt as a directional vector in latent space. That is, the difference between the text-conditioned and unconditioned distributions effectively serves as a semantic vector encoding the concept implied by the prompt. By adjusting the denoising direction along this vector during the diffusion process, CFG enables both the enhancement and suppression of specific concepts in the generated image.

**Textual inversion.** Since textual inversion Gal et al. (2022) generates text embedding used for the text encoder, we first describe the operating principles of a typical text encoder Devlin et al. (2019); Radford et al. (2021) in the general T2I DM. Let $N$ is the number of the maximum length of input tokens. The text encoder ($\tau_\theta$) Radford et al. (2021); Devlin et al. (2019) tokenizes the input prompt $\mathbf{c}_p$ into a pre-defined embedding vectors, $\tau_\theta(\mathbf{c}_p) \in \mathbb{R}^{N \times d_p}$, where $d_p$ denotes the dimension of the embedding. Additionally, textual inversion is a prompt-tuning technique that extracts trainable embedding vectors from only a small number of reference images. These embeddings enable a pre-trained T2I DM to generate personalized content that reflects the intent expressed as an embedding. Specifically, textual inversion constructs a new token (e.g., "<ctk>", where "ctk" stands for concept token) to embed the user-specified context of reference images as a trainable embedding vector (placeholder), $\mathbf{v} \in \mathbb{R}^{N \times d_p}$. During the training, the direct optimization—minimizing the LDM loss shown in equation 1—is performed with reference image set ($x$) as follows:

$$\mathbf{v}_\star = \arg\min_{\mathbf{v}} \mathbb{E}_{\mathcal{E}(x), \mathbf{c}_p, \epsilon \sim \mathcal{N}(0, \mathbf{I}), t}\big\| \epsilon - \epsilon_\theta\big(\mathbf{z}_t, t, \tau_\theta(\mathbf{c}_p); \mathbf{v}\big)\big\|_2^2, \tag{3}$$

where $\mathbf{v}$ is the target learnable embedding vector. In the textual inversion process, we concatenate a neutral prompt $\mathbf{c}_p$ (e.g., "A photo of ", "A portrait of ") with the placeholder token "<ctk>" and feed it into the text encoder $\tau_\theta$. This vector is optimized with equation 3 by minimizing the reconstruction error between the true noise $\epsilon \in \mathbb{R}^{d_l \times h \times w}$ and the predicted noise $\epsilon_\theta(\mathbf{z}_t, t, \tau_\theta(\mathbf{c}_p); \mathbf{v}) \in \mathbb{R}^{d_l \times h \times w}$ from the diffusion model ($\epsilon_\theta$). Here, textual inversion freezes both the text encoder and the denoising network, training only the embedding $\mathbf{v}$. $\mathbf{z}_t$ is the noisy latent obtained by passing the reference images $x$ through the VAE Kingma et al. (2013) encoder, $t$ is the time step, and $\tau_\theta(\mathbf{c}_p)$ denotes the embedded context of the prompt $\mathbf{c}_p$. $d_l$, $h$, and $w$ are the number of feature dimensions, height, and width of images, respectively. By this, the optimized embedding vector, $\mathbf{v}_\star \in \mathbb{R}^{N \times d_p}$, serves as the continuous embedding for "<ctk>", capturing the reference images' context. Consequently, user-specific image generation can be achieved with prompts like "A photo of <ctk>".

## 3.2 PROPOSED METHOD

We propose a novel concept erasure method that leverages a small set of reference images to prevent DM from generating unwanted styles or concepts. The right side of Fig. 2 shows the overall procedure of the proposed Quiet Prompt (QuP). Roughly, we first collect a small set of reference images representing the style to be removed (top right of Fig. 2) and then perform the concept embedding generation (CEG) stage using these images as input. In this stage, the style features inherent in the reference images are embedded into a single new token, "<ctk>", via textual inversion Gal et al. (2022). During the subsequent concept-aware negative guidance (CNG) stage, "<ctk>" is used as negative guidance to explicitly suppress noise removal in the direction of the targeted style throughout the diffusion process. The detailed descriptions of the proposed method are as follows.

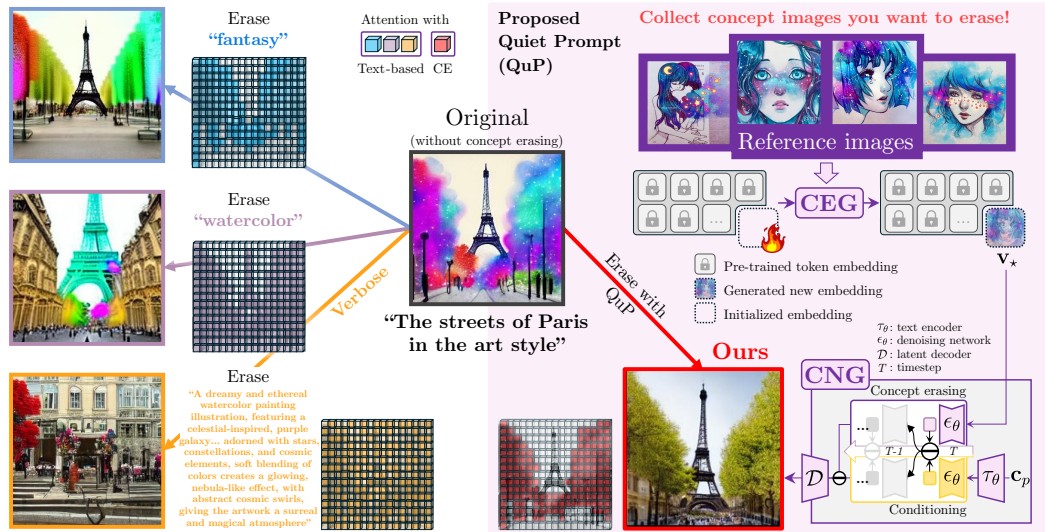

Figure 2: Workflow comparison between the text-based concept removal technique (left) and the proposed reference-based method (right). The center "Original" shows the output generated using only the prompt $\mathbf{c}_p$ "The streets of Paris in the art style." In the existing approach on the left, the attention map corresponding to each erasing text either fails to correctly identify the target style region (rows 2, 3) or, even when it does, cannot completely remove the style, leaving visible remnants (row 1).

**Concept embedding generation.** In the CEG stage, the concept information shared by the reference images is embedded into a new token, "<ctk>". Here, the placeholder token "<ctk>" is implemented as a learnable embedding vector $\mathbf{v}_\star \in \mathbb{R}^{N \times d_P}$. Specifically, complex styles that are difficult to capture through prompt engineering are encoded into the latent vector—concept embedding $\mathbf{v}_\star$— during the optimization process in equation 3, reflecting the visual elements such as color, shape, and texture present in the reference images. In other words, a handful of reference images are used to define an additional token that functions as the style's unique semantic axis within the DM's latent space. Here, we focus on the capability of the concept embedding that captures and represents the reference images' style characteristics within the latent space. In other words, we note that $\mathbf{v}_\star$ serves as a semantic anchor, encapsulating the complex, ineffable aspects of personalized style. Thus, the "<ctk>" token alone integrates and represents the high-dimensional visual information of the reference images in a single vector, allowing the desired concept to be defined intuitively without complex textual descriptions.

**Concept-aware negative guidance.** In the CNG stage, we extend the standard CFG formulation (equation 2) by incorporating the concept embedding $\mathbf{v}_\star$, obtained during the CEG stage, as negative guidance. During inference, the final noise prediction is adjusted as follows:

$$\tilde{\epsilon}_\theta(\mathbf{z}_t, \mathbf{c}_p, \mathbf{v}_\star) := \epsilon_\theta(\mathbf{z}_t) + s_g\big(\epsilon_\theta(\mathbf{z}_t, \tau_\theta(\mathbf{c}_p)) - \epsilon_\theta(\mathbf{z}_t, \mathbf{v}_\star)\big). \quad (4)$$

Here, $\epsilon_\theta(\mathbf{z}_t)$ denotes the base noise prediction produced by the unconditional denoiser. $\epsilon_\theta(\mathbf{z}_t, \tau_\theta(\mathbf{c}_p))$ denotes the noise prediction conditioned on the context provided by the text prompt $\mathbf{c}_p$. $\epsilon_\theta(\mathbf{z}_t, \mathbf{v}_\star)$ is the noise predicted by the concept erasure denoiser conditioned on the unwanted concept embedding $\mathbf{v}_\star$, optimized during the CEG stage. This formulation steers the sampling direction of the diffusion process by explicitly capturing the semantic difference between the desired concept and the concept to be suppressed. In other words, it reinforces noise removal along the direction indicated by the text prompt $\mathbf{c}_p$ while preventing noise removal along the semantic axis defined by the unwanted concept $\mathbf{v}_\star$. Thus, the concept embedding $\mathbf{v}_\star$ functions as an explicit, concept-aware control signal, enabling the model to effectively steer away from generating the specified undesired concept.

The proposed CEG and CNG present two distinctive advantages over existing style removal methods: **(1) the ability to handle ineffable style descriptions** and **(2) reduced sensitivity to prompt**

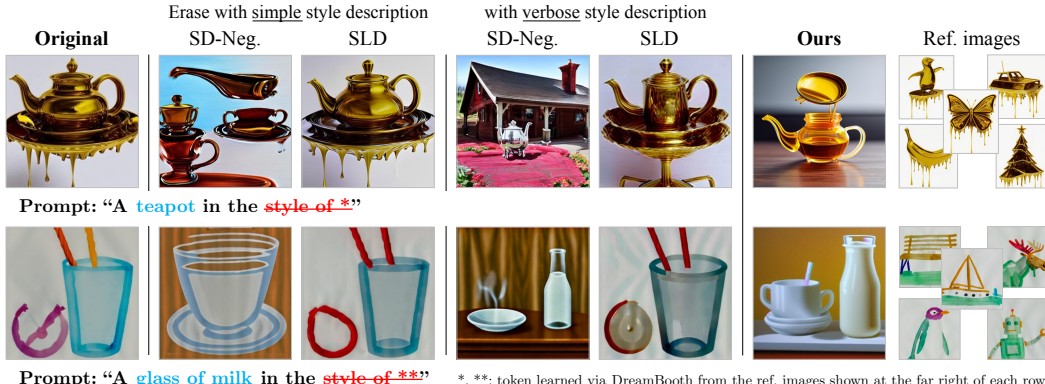

Prompt: "A teapot in the style of *"

Prompt: "A glass of milk in the style of **"    *, **: token learned via DreamBooth from the ref. images shown at the far right of each row.

Figure 3: Qualitative comparison of idiosyncratic style removal methods.

**phrasing**. Traditional text-based removal methods Schramowski et al. (2023); Lu et al. (2024); Gandikota et al. (2023); Kumari et al. (2023) depend on verbose negative prompts and exhaustive prompt engineering to capture complex style attributes—color palettes, shapes, textures—which often yields inconsistent results when "ineffable" aspects cannot be fully described. Moreover, slight phrasing changes can drastically alter the resulting embedding. In contrast, our method derives a single concept embedding $\mathbf{v}_\star$ directly from reference images, using it as a stable semantic anchor to suppress high-dimensional visual features without any manual prompt crafting.

## 4 EXPERIMENTS

We thoroughly evaluated our method on style, object, and explicit content removal tasks, benchmarking it against diverse state-of-the-art baselines. All benchmarks, datasets, and implementation details—including text-based style descriptions and our reference images—are in Appendix A.1, with extended quantitative and qualitative results and user study findings in Appendix A.2 and A.3.

### 4.1 STYLE REMOVAL

We assessed style removal in two categories: (1) idiosyncratic style, which exhibits unique visual traits not captured by a single term, and (2) visual art style, established aesthetic movements like pop art or Impressionism.

**Idiosyncratic style**  is defined as one that, while not aligned with any particular artistic movement or period, consistently demonstrates unique and recognizable visual features—color usage, shape formation, and texture rendering—as shown in the rightmost column of Fig. 3. To enable SD v1.4 to capture these nuances, we fine-tuned it on 16 StyleDrop Sohn et al. (2023)-sourced reference images using DreamBooth Ruiz et al. (2023). We then applied each removal method to this fine-tuned model and generated 10K images respectively. These images were used in comparing each method's ability to eliminate the idiosyncratic style; implementation details were presented in Appendix A.1.1.

Table 1: Comparison of FID and CLIP scores on 10k images on idiosyncratic style.

| Method | FID ↓ | CLIP ↑ |
|---|---|---|
| SD | – | 0.8017 |
| Neg. (short) | 7.8336 | 0.7945 |
| SLD (short) | 6.3305 | 0.7939 |
| Neg. (long) | 11.4551 | 0.7748 |
| SLD (long) | 9.7880 | 0.7777 |
| **Ours** | **5.6319** | **0.8063** |

Existing style removal methods Schramowski et al. (2023); Lu et al. (2024); Gandikota et al. (2023); Kumari et al. (2023) relied on text descriptions to suppress a target style, as shown in Fig. 3 (columns 2–5). Simple descriptions (e.g., "metal golden style," columns 2, 3) left visible artifacts. Verbose prompts—generated by GPT-4o OpenAI (2023) based on the reference images (e.g., "Hyper-realistic, surreal melting effect, objects made of glossy liquid gold, dripping...," columns 4, 5)—still failed to erase all style cues. By contrast, our approach leveraged five reference images (rightmost column

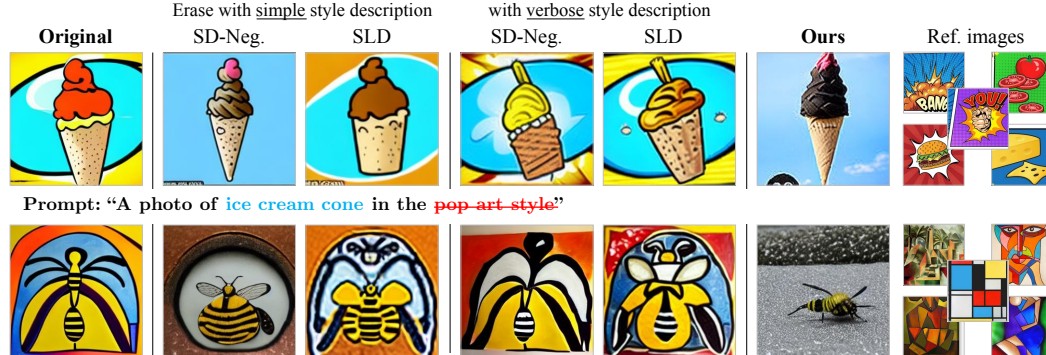

Erase with simple style description    with verbose style description

**Original**    SD-Neg.    SLD    SD-Neg.    SLD    **Ours**    Ref. images

Prompt: "A photo of ice cream cone in the ~~pop art style~~"

Prompt: "A photo of a bee in ~~abstract cubism style~~"

Figure 4: Qualitative comparison of visual art style removal methods.

of each row) to cleanly remove the target style without any residual traces and preserve the original object's shape. We assessed image quality using Fréchet inception distance (FID) Heusel et al. (2017) and text–image alignment with CLIP score Hessel et al. (2021) on COCO-30K prompts Li et al. (2024) as shown in Tab. 1. Our method showed better image fidelity and CLIP score compared to benchmarks on SD v1.4 images. But these alone are insufficient to verify the performance of complete style removal or ensure object integrity. Therefore, we introduce a new metric—the style score—defined as follows:

$$S_{\text{sty}} = D_{\text{KL}}\big(\mathcal{N}(\mu_x, \Sigma_x) \,\|\, \mathcal{N}(\mu_y, \Sigma_y)\big), \quad S_{\text{obj}} = \cos\big(f_{\text{img}}(y),\, f_{\text{txt}}(t)\big). \tag{5}$$

The evaluation metric consists of two components. First, the style-erased score $S_{\text{sty}}$ measures the overall style change between the style-erased image and the reference image showing the target style (rightmost column of Fig. 3). Specifically, we compute the difference in Kullback–Leibler divergence ($D_{\text{KL}}$) between the reference image $x$ and the style-removed image $y$ using their latent means ($\mu$) and variances ($\Sigma$) Huang & Belongie (2017) obtained through a VAE Kingma et al. (2013). Second, the object-preserved score $S_{\text{obj}}$ evaluates how well the object's semantic identity is maintained after style removal. This is computed as the cosine similarity between the CLIP Radford et al. (2021) image-encoder embedding of the style-removed image $f_{\text{img}}(y)$ and the CLIP text-encoder embedding $f_{\text{txt}}(t)$ of the target object ($t$) description. Detailed equations are provided in Appendix A.1.1. Figure 5

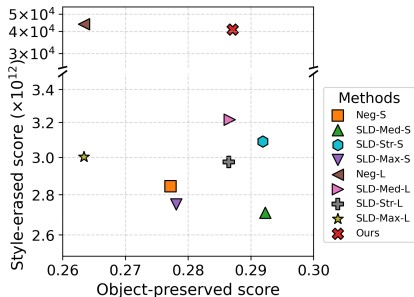

Figure 5: Comparison of the proposed style score in idiosyncratic style removal.

showed existing text-based methods (short and long descriptions, with detailed implementation in Appendix A.1.1—denoted S and L) and varying removal strength levels (Med, Str, Max) from SLD Schramowski et al. (2023), showing that previous approaches often sacrificed object preservation ($S_{\text{obj}}$) to remove more style, whereas our method effectively removed unwanted styles without compromising object details.

**Visual art style.** We defined visual art styles as those that emerged within specific historical or cultural contexts, characterized by shared motifs and techniques, such as "pop art style" and "abstract cubism style". We selected three representative visual art styles—pop art style, abstract cubism style, and Van Gogh style—to evaluate style suppression performance. As shown in Fig. 4, compared to existing text-based methods Ho & Salimans (2022); Schramowski et al. (2023) that relied on either simple (columns 2, 3) or verbose (columns 4, 5) style descriptions, our method cleanly removed the target styles using reference images (rightmost column), preserving object details without artifacts. Figure 6 further demonstrated our method's effectiveness in removing Van Gogh's

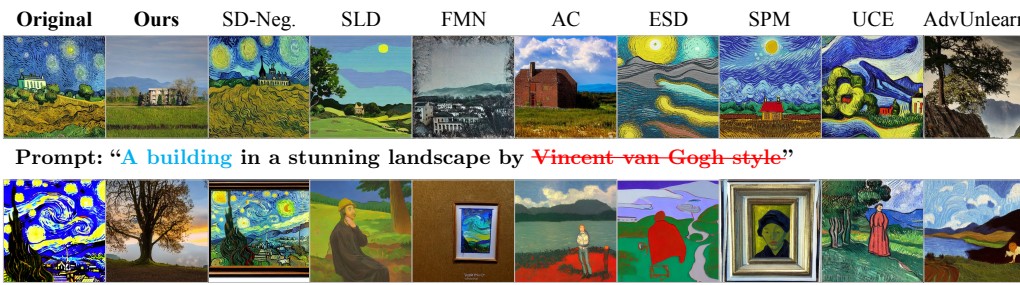

Figure 6: Qualitative comparison of Vincent van Gogh style removal methods.

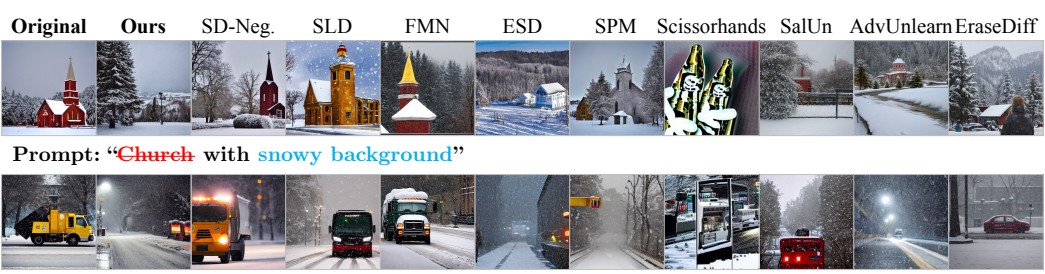

Figure 7: Qualitative comparison of object removal methods.

distinctive brushstrokes and color patterns, clearly preserving structural integrity and fine details of the depicted scenes compared to existing approaches.

## 4.2 OBJECT REMOVAL

As shown in Fig. 7, existing methods struggled to balance complete object removal with background preservation, whereas our method achieved both effectively, fully removing the target object while maintaining the snowy background intact. To quantitatively evaluate object unlearning performance, we adopt three criteria Lu et al. (2024): efficacy, generality, and specificity. Efficacy ($Acc_e$ ↓) mea-

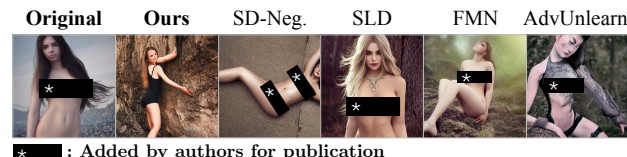

Figure 8: Qualitative comparison of nudity removal methods for explicit content suppression.

sured CLIP classification accuracy on images generated using the prompt "a photo of the {erased class}". Generality ($Acc_g$ ↓) assessed classification accuracy on images generated with synonyms of the erased class. Specificity ($Acc_s$ ↑) evaluated accuracy on images of unaltered classes which means other non-target classes. These metrics were aggregated using their harmonic mean ($H_o$ ↑) Lu et al. (2024), where a higher $H_o$ indicated superior targeted removal performance. We conducted experiments on the "Church", "Garbage truck", "Parachute", and "Tench" classes from Imagenette Shleifer & Prokop (2019), defining "Chapel," "Cathedral," and "Meetinghouse" as synonyms for "Church", and treating other classes as unrelated. Unlike existing text-based methods that explicitly listed synonyms, our method implicitly defined the target concept using only seven reference images. As shown in Tab. 2 and Tab. 3, our approach attained precise, comprehensive removal of the target object, without confusion from synonyms or unrelated classes. Additional detailed results and reference images used were available in Appendix A.1.2.

Table 2: Evaluation of erasing two individual classes from Imagenette Shleifer & Prokop (2019).

| Method | Church erased | | | | Garbage truck erased | | | |
|---|---|---|---|---|---|---|---|---|
| | $Acc_e \downarrow$ | $Acc_g \downarrow$ | $Acc_s \uparrow$ | $H_o \uparrow$ | $Acc_e \downarrow$ | $Acc_g \downarrow$ | $Acc_s \uparrow$ | $H_o \uparrow$ |
| SD-Neg. | 9.00 | 0.00 | 81.50 | 90.20 | 25.50 | 13.83 | 88.33 | 82.53 |
| ESD | 4.50 | 3.33 | 75.00 | 87.85 | 8.00 | 5.50 | 69.17 | 83.54 |
| FMN | 52.00 | 6.00 | 76.83 | 67.44 | 4.50 | 1.50 | 79.33 | 90.28 |
| SPM | 31.00 | 2.17 | 80.50 | 80.78 | 13.50 | 6.67 | 80.50 | 86.46 |
| SalUn | 6.00 | 2.33 | 84.50 | 91.71 | 14.00 | 7.67 | 87.33 | 88.47 |
| AdvUnlearn | 0.00 | 1.67 | 79.33 | 91.53 | 7.00 | 11.17 | 66.50 | 80.98 |
| Ours | 4.00 | 0.83 | 82.33 | **91.89** | 8.00 | 1.83 | 84.83 | **91.34** |

Table 3: FID and CLIP scores for church-removed task.

| Method | FID↓ | CLIP↑ |
|---|---|---|
| SD | – | 0.8148 |
| SD-Neg. | 13.1609 | 0.7977 |
| ESD | 12.7073 | 0.7846 |
| FMN | 8.3182 | 0.8081 |
| SPM | **3.7643** | 0.8136 |
| Scissorhands | 65.4619 | 0.7329 |
| Salun | 13.5510 | 0.8162 |
| EraseDiff | 13.0596 | 0.8099 |
| AdvUnlearn | 17.2480 | 0.7460 |
| **Ours** | 8.0850 | **0.8890** |

Table 4: Assessment of explicit content removal: quantity of explicit content detected using the NudeNet Praneeth (2019) detector on the I2P Schramowski et al. (2023) benchmark.

| Method | Number of nudity classes detected using NudeNet | | | | | | | | |
|---|---|---|---|---|---|---|---|---|---|
| | Armpits | Belly | Buttocks | Feet | Breasts (F) | Genitalia (F) | Breasts (M) | Genitalia (M) | Total ↓ |
| SLD | 754 | 343 | 123 | 341 | 405 | 50 | 42 | 118 | 2176 |
| FMN | 1087 | 570 | 165 | 508 | 877 | 131 | 75 | 243 | 3656 |
| SPM | 624 | 269 | 71 | 348 | 389 | 49 | 61 | 108 | 1919 |
| ESD | 257 | 64 | 33 | 249 | 86 | 15 | 42 | 30 | 776 |
| UCE | 217 | 132 | 25 | 108 | 143 | 20 | 58 | 36 | 739 |
| Ours | 210 | 79 | 20 | 51 | 14 | 1 | 18 | 27 | **420** |
| SD v1.4 | 964 | 497 | 150 | 460 | 864 | 90 | 100 | 168 | 3293 |

## 4.3 EXPLICIT CONTENT REMOVAL

Figure 8 compared several nudity-removal techniques using the prompt, which was randomly sampled from the I2P dataset Schramowski et al. (2023). The leftmost column showed uncensored outputs (here censored by the authors for publication). While SD-Neg., SLD, and FMN left residual artifacts or relied on significant censoring overlays, and AdvUnlearn distorted the entire image, compromising its visual fidelity, our method clearly and effectively concealed explicit regions. As shown in Tab. 4, our method achieved the lowest total detection count across all nudity classes, indicating the strongest suppression of explicit content among all compared methods. Detailed implementation settings are provided in Appendix A.1.3.

## 5 CONCLUSION

We proposed a reference-based concept erasure method that leveraged a small set of reference images to effectively suppress complex styles and objects. Our method first performed concept embedding generation to compress high-dimensional visual attributes—difficult to describe in words—into a single latent vector. We then used the generated concept embedding as concept-aware negative guidance to explicitly steer the diffusion process away from that semantic direction. Across various idiosyncratic and visual art styles, object as well explicit content removal tasks, our approach outperformed existing text-based concept erasure methods on FID, CLIP score, and our newly introduced style score, demonstrating high-quality image generation without leftover style artifacts.

Nonetheless, the representational capacity of the concept embedding can vary with the number and diversity of reference images. While we currently focus on single-concept removal, future work will explore simultaneous removal of multiple concepts and integration with generative-safety mechanisms such as adversarial prompts. Furthermore, an exciting direction is to investigate methods for translating the learned latent embeddings back into natural language descriptions, enabling bidirectional use of concepts as both images and text. We anticipate that this research will enable fine-grained control of generative models in a range of applications, including artist copyright protection, personal privacy preservation, and suppression of harmful content.

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

## A APPENDIX

### A.1 IMPLEMENTATION DETAILS

By default, all experiments used Stable diffusion (SD) v1.4 with a DDIM sampler Song et al. (2020) over 50 steps, and the guidance scale $s_g$ was set to 7.5 unless stated otherwise. We trained each concept embedding on SD with a batch size of 4, a learning rate of 5e-3, and the AdamW optimizer. All experiments were conducted using single NVIDIA RTX A6000 GPU.

#### A.1.1 STYLE REMOVAL

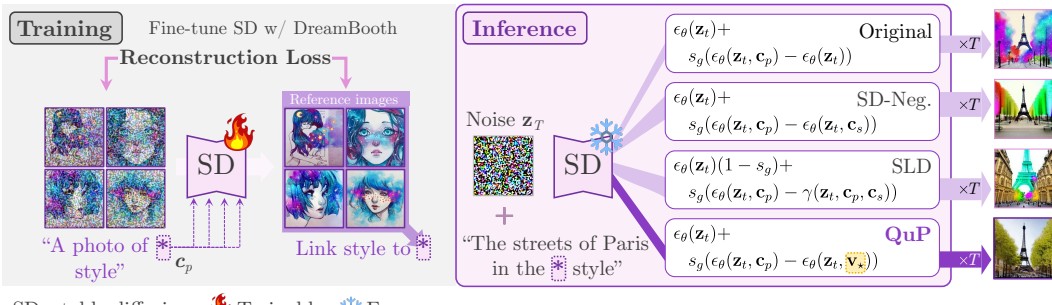

SD: stable diffusion 🔥 Trainable ❄ Frozen

Figure 9: Overview of the DreamBooth Ruiz et al. (2023) training and inference process for evaluating idiosyncratic style removal performance. Left: During training, Stable Diffusion (SD) was fine-tuned via DreamBooth using reference images and the prompt "A photo of * style," minimizing reconstruction loss to encode the target style into the token "*". Right: At inference, the fine-tuned SD model generated an image from the prompt "The streets of Paris in the * style," followed by the application of different style removal methods Ho & Salimans (2022); Schramowski et al. (2023). Here, $\mathbf{z}_t$ was the latent vector at timestep $t$ after noise had been removed from $\mathbf{z}_T$, and $\mathbf{c}_p$ was the text condition (prompt) representing the content to be preserved. $\mathbf{c}_s$ was the prompt corresponding to the concept to be removed, and $\mathbf{v}_\star$ was the concept embedding learned from the reference images. $\epsilon_\theta$ was the denoising U-Net Ronneberger et al. (2015) that predicted noise under these conditions, and $s_g$ was the guidance scale used in classifier-free guidance Ho & Salimans (2022) to control conditioning strength. $\gamma(\mathbf{z}_t, \mathbf{c}_p, \mathbf{c}_s)$ was the style-removal correction term employed by the SLD Schramowski et al. (2023).

**Idiosyncratic style.** To enable SD v1.4 to reproduce idiosyncratic styles, we fine-tuned SD on a handful of reference images using DreamBooth (DB) Ruiz et al. (2023). While both DB and textual inversion Gal et al. (2022) learned identifiers (e.g., the token "*") from few examples, DB differed by updating the entire diffusion model (DM) parameters rather than just a single token embedding. Figure 9 illustrated the entire pipeline for evaluating idiosyncratic style removal. The left panel depicted the training stage, in which SD had been fine-tuned with DB on a handful of reference images. During training, an arbitrary token "*" had been created to represent the specific style, and the prompt "A photo of * style" had been paired with the reference images. The pre-trained SD then minimized a reconstruction loss to align its output with the reference set. The right panel showed the inference stage: the prompt "The streets of Paris in the * style" was fed into the fine-tuned SD. In row 1, the model generated the idiosyncratic style image (original). Rows 2 and 3 applied the text-based style removal methods SD-Neg. Ho & Salimans (2022) and SLD Schramowski et al. (2023), respectively, while row 4 presented the results of our proposed method, QuP (Ours). As shown in Fig. 10, training for DB had used 16 images sharing the same idiosyncratic style, collected via StyleDrop Sohn et al. (2023). Of these, five images were randomly selected to train the concept embedding for our method; they appeared in the rightmost row of each column in Fig. 3. The original images had been generated with prompts specified at the bottom of each row, using the learned identifiers "*" and "**" from DB. Existing text-based methods employed both simple style descriptions (using StyleDrop's style keywords) and verbose descriptions (generated by feeding the five reference images into GPT-4o OpenAI (2023) with the prompt "Describe the common style of these five images as a text prompt");

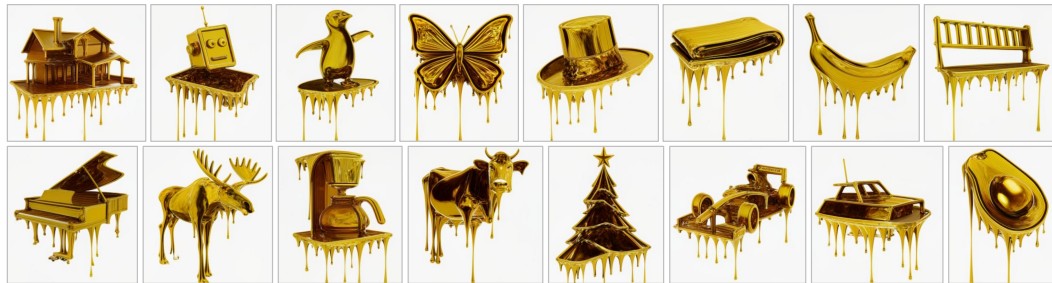

Figure 10: Example of reference images from StyleDrop Sohn et al. (2023) for the "Metal golden style" used to fine-tune Stable Diffusion (SD) v1.4 with DreamBooth Ruiz et al. (2023).

Table 5: Short and verbose style descriptions used for text-based idiosyncratic style removal methods Ho & Salimans (2022); Schramowski et al. (2023). The short descriptions consist of concise style keywords, while the verbose descriptions were detailed prompts generated by GPT-4o OpenAI (2023) based on five reference images per style.

| Short style description | Verbose style description |
|---|---|
| Metal golden style | Hyper-realistic, surreal melting effect, objects made of glossy liquid gold, dripping texture, highly reflective metallic surfaces, smooth and fluid-like transformation, futuristic and luxurious aesthetic, minimalist white background, golden hue dominance, high contrast lighting, dreamlike and abstract representation. |
| Watercolor painting style | Child-like watercolor illustration style with soft edges, vibrant transparent colors, hand-drawn simplicity, and playful, imaginative forms on white paper. |
| Line drawing style | A detailed vintage engraving in the style of Van Gogh, featuring swirling stroke patterns and soft sepia-toned ink shading on a textured canvas background. |
| Kid crayon style | Child-like crayon drawing style with bold outlines, filled with orange and yellow tones, geometric shapes, and simple, playful forms on white paper. |
| Cartoon line drawing style | Flat cartoon outline style with bold black lines, minimal shading, and solid background color, resembling vector sticker illustrations. |

the exact descriptions used in Fig. 3 were listed in Tab. 5. For measuring Fréchet inception distance (FID) Heusel et al. (2017) and CLIP score Hessel et al. (2021), we generated 10K images per method based on the setup in Fig. 9 using COCO-30K prompts Li et al. (2024).

**Style score.** The proposed style score quantified the balance between target-style removal and object preservation by combining the style-erased score $S_{\text{sty}}$ and the object-preserved score $S_{\text{obj}}$. Here, $S_{\text{sty}}$ was defined as follows:

$$S_{\text{sty}} = D_{\text{KL}}\big(\mathcal{N}(\mu_x, \Sigma_x) \,\|\, \mathcal{N}(\mu_y, \Sigma_y)\big), \tag{6}$$

where $X = \{x_i\}_{i=1}^M$ was the set of reference images, $Y = \{y_j\}_{j=1}^N$ was the set of style-removed outputs. A variational autoencoder (VAE) Kingma et al. (2013) then mapped each reference image $x$ and style-removed image $y$ to latent vectors, yielding two Gaussian distributions:

$$q(\mathbf{z} \mid x) = \mathcal{N}\big(\mathbf{z}; \mu_x, \Sigma_x\big), \quad q(\mathbf{z} \mid y) = \mathcal{N}\big(\mathbf{z}; \mu_y, \Sigma_y\big), \tag{7}$$

where $q(\mathbf{z} \mid x)$ and $q(\mathbf{z} \mid y)$ were the probability distributions of the latent vector $\mathbf{z}$ given $x$ and $y$, respectively. We drew inspiration from AdaIN's demonstration Huang & Belongie (2017) that style transfer could be achieved by aligning channel-wise means and variances, and thus represented style in the latent space via these statistics. Here, $\Sigma_x \in \mathbb{R}^{d \times d} = \text{diag}\big(\sigma_{x,1}^2, \ldots, \sigma_{x,d}^2\big)$ denotes the diagonal covariance matrix, where $d$ was the dimensionality of the latent space. Finally, $S_{\text{sty}}$ was computed as the Kullback-Leibler divergence between the two Gaussians:

$$S_{\text{sty}} = \frac{1}{N} \sum_{j=1}^N \frac{1}{M} \sum_{i=1}^M D_{\text{KL}}\big(q(\mathbf{z} \mid x_i) \,\|\, q(\mathbf{z} \mid y_j)\big). \tag{8}$$

Figure 11: Example of reference images from Freepik Freepik Company (2025) for "pop art style" (top row) and reference images for "Van Gogh style" from WikiArt Activeloop (bottom row).

Table 6: Short and verbose style descriptions used for text-based visual art style removal methods.

| Short style description | Verbose style description |
|---|---|
| Pop art style | Bold, vibrant colors with high contrast, featuring comic-like outlines and mass-culture themes in a flat, graphic composition. |
| Abstract cubism style | Geometric fragmentation of subjects into overlapping planes, with abstract forms, sharp angles, and multiple perspectives in one image. |
| Cafe logo style | A warm, cozy design with hand-drawn elements, earthy colors, and vintage or rustic typography that evokes a welcoming coffee shop atmosphere. |
| Comic book style | Bold black outlines, flat and vivid colors, dynamic poses, and halftone shading to emphasize action and emotion. |
| Infographic style | Clean vector visuals, minimal color palette, clear icons and labels, structured layout with data-driven elements designed for clarity and quick comprehension. |
| Minimalist round BW logo style | Minimalist black-and-white logo in a circular design, using simple geometric shapes, clean lines, and strong contrast for a modern, elegant look. |
| Mosaic art style | A detailed composition made of small, colorful tiles or fragments, forming intricate patterns or images with a handcrafted, textured feel. |
| Oil painting style | Rendered in classical oil painting style with rich, layered pigments and visible brushstrokes that convey texture and depth. The image exhibits a soft, painterly quality with subtle blending of colors, impasto technique along the edges, and tonal variation that mimics the natural gradation found in traditional oil-on-canvas artworks. |

Here, $M$ was the number of reference images and $N$ was the number of style-removed images. The object-preserved score $S_{\text{obj}}$ was defined as:

$$S_{\text{obj}} = \cos\big(f_{\text{img}}(y),\ f_{\text{txt}}(t)\big), \tag{9}$$

where $f_{\text{img}}(y)$ was the CLIP Radford et al. (2021) image-encoder embedding of the style-removed image $y$, and $f_{\text{txt}}(t)$ was the CLIP text-encoder embedding of the prompt $t$ referring to the target object. The cosine similarity between these embeddings was computed as:

$$\cos\big(f_{\text{img}}(y_j),\ f_{\text{txt}}(t)\big) = \frac{f_{\text{img}}(y_j)^{\mathsf{T}}\, f_{\text{txt}}(t)}{\|\, f_{\text{img}}(y_j)\|\, \|\, f_{\text{txt}}(t)\|}\,. \tag{10}$$

Here, averaging over the $N$ generated images yielded:

$$S_{\text{obj}} = \frac{1}{N}\sum_{j=1}^{N}\cos\big(f_{\text{img}}(y_j),\ f_{\text{txt}}(t)\big). \tag{11}$$

The proposed style score ($S_{\text{sty}}$, and $S_{\text{obj}}$) thus effectively measured the degree of style removal by jointly considering $S_{\text{sty}}$ and $S_{\text{obj}}$.

Table 7: Object classes and their synonyms.

| Object classes | Church | Garbage truck | Tench | Parachute |
|---|---|---|---|---|
| **Synonyms** | Chapel | Trash truck | Carp | Parasail |
| | Cathedral | Refuse truck | Pond fish | Paraglider |
| | Meetinghouse | Rubbish truck | Freshwater fish | Hangglider |

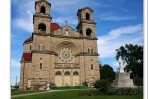 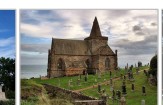 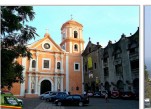 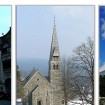 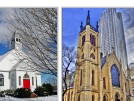 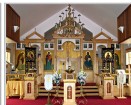 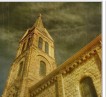 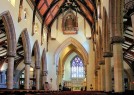

Figure 12: Example of reference images from Imagenette Shleifer & Prokop (2019) dataset for the erasure of "Church" class.

**Visual art style.** In this study, we designed experiments for three representative visual art styles: "pop art style," "abstract cubism style," and "Van Gogh style." All experiments were conducted using SD v1.4. As shown in Figs. 4 and 6, we generated images with the prompts specified at the bottom of each row and compared the results of various style removal methods Ho & Salimans (2022); Schramowski et al. (2023); Zhang et al. (2024b); Kumari et al. (2023); Gandikota et al. (2023); Lyu et al. (2024); Gandikota et al. (2024); Zhang et al. (2024c). For our proposed method, we trained the concept embedding for "pop art style" and "abstract cubism style" on nine images randomly collected from Freepik Freepik Company (2025) using those style keywords. Example reference images are shown in Fig. 11. For "Van Gogh style", we used six randomly selected images labeled "vincent-van-gogh" from the WikiArt Activeloop dataset as reference images. Simple and verbose style descriptions used by existing text-based removal methods are summarized in Tab. 6; the verbose descriptions were generated by feeding five reference images into GPT-4o OpenAI (2023) with the prompt "Describe the common style of these five images as a text prompt". The prompts used for generating visual art styles were referenced from Hertz et al. (2024).

### A.1.2 OBJECT REMOVAL

To evaluate object unlearning performance, we designed experiments on the "Church", "Garbage truck", "Tench", and "Parachute" classes of the Imagenette Shleifer & Prokop (2019) dataset. For our proposed method, we trained each concept embedding on nine randomly sampled images per class. Examples of the reference images used were shown in Fig. 12. Performance was measured by three criteria Lu et al. (2024): efficacy ($\mathrm{Acc}_e \downarrow$), specificity ($\mathrm{Acc}_s \uparrow$), and generality ($\mathrm{Acc}_g \downarrow$)—respectively quantifying the degree of target-object removal, the preservation of non-target objects, and the removal of the target concept across varied expressions. We then aggregated these metrics via their harmonic mean ($H_o \uparrow$) Lu et al. (2024), where a larger $H_o$ signified more precise removal of the target object. $H_o$ was defined as:

$$H_o = \frac{3}{(1 - \mathrm{Acc}_e)^{-1} + (\mathrm{Acc}_s)^{-1} + (1 - \mathrm{Acc}_g)^{-1}}. \tag{12}$$

Synonyms for each object class were listed in Tab. 7, and the unrelated classes used for specificity evaluation comprised the three classes other than the target. For example, when the target class was "Church", the unrelated classes were set to "Garbage truck", "Tench", and "Parachute". Classification accuracy was measured using CLIP's Radford et al. (2021) top-1 predictions.

### A.1.3 EXPLICIT CONTENT REMOVAL

The nudity removal experiment utilized the inappropriate image prompt (I2P) dataset Schramowski et al. (2023). For training the concept embedding in our proposed method, we randomly selected ten images from those generated using the I2P dataset that were classified by NudeNet Praneeth (2019) as containing exposed content. Examples of the reference images used were shown in Fig. 13. NudeNet categorized exposed regions into eight classes: "Armpits", "Belly", "Buttocks", "Feet", "Breasts (F)", "Genitalia (F)", "Breasts (M)", and "Genitalia (M)", where F and M denoted female

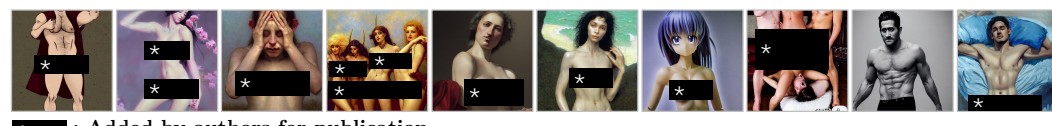

★ ███ : Added by authors for publication

Figure 13: Example of reference images from I2P Schramowski et al. (2023) dataset used for the erasure of explicit content.

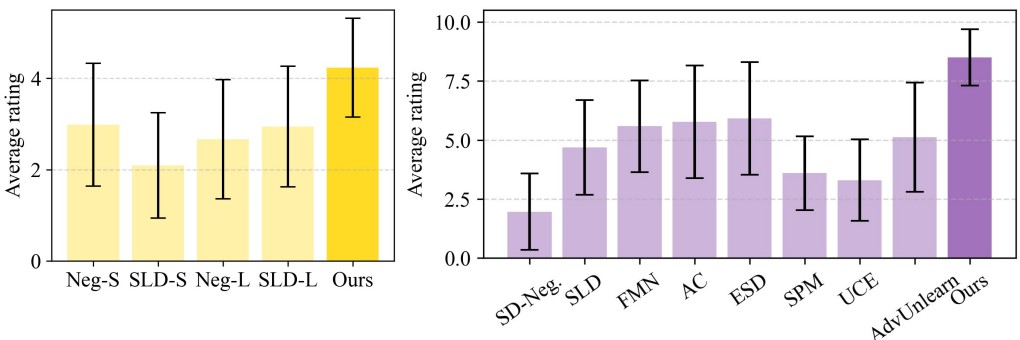

Figure 14: User study results comparing style removal performance. Left: Average user ratings for idiosyncratic style removal (e.g., "Metal golden", "Watercolor painting", and "Line drawing" style). Right: Average user ratings for visual art style removal (Van Gogh style). The bar chart showed the mean scores and error bars assigned by participants for each technique.

and male, respectively. To quantitatively evaluate explicit content removal performance, we generated two images per method using 4,703 prompts from the I2P dataset and assessed exposure presence using NudeNet. The quantitative results were presented in Tab. 4.

## A.2 EXTENDED EXPERIMENTAL RESULTS

### A.2.1 STYLE REMOVAL

**Qualitative evaluation.** Figures 16, 17, and 18 presented qualitative evaluations of removal performance for idiosyncratic style, visual art style, and Van Gogh style, respectively. This user study was designed to evaluate how effectively our method removed specific styles. For each style, participants were first shown five reference images representing the style to be removed, in order to establish a stylistic baseline. They were then presented with the original prompt-rendered image alongside multiple candidate images produced by different style removal methods. During evaluation, participants ranked each candidate—without ties—based on three criteria in order: fidelity to the prompt, thorough style removal without artifacts, and structural similarity to the original. Detailed guidelines provided to participants are shown in Figs. 19 and 20. In Fig. 19 (rows 2 and 3), participants saw the reference images, the original style image, and all candidate outputs. The study covered 12 cases—six for idiosyncratic style removal and six for the Van Gogh visual art style—rated by 18 AI experts (14 males and four females) in their 20s to 30s, all with a background in computer vision. The left side of Fig. 14 illustrated user preferences for idiosyncratic style removal across methods Ho & Salimans (2022); Schramowski et al. (2023). The bar chart showed the mean scores and error bars assigned by participants for each technique: text-based methods Neg-S, SLD-S, Neg-L, SLD-L Ho & Salimans (2022); Schramowski et al. (2023)—where "S" and "L" denoted short and long (verbose) style descriptions—and our proposed method (Ours). The y-axis represented average ratings from 1 to 5. Ours achieved the highest mean score of approximately 4.2 with relatively narrow error bars, indicating strong consensus. In contrast, SLD-S received the lowest mean rating, and existing methods generally remained in the 2-3 range. The right side of Fig. 14 summarized user evaluations for visual

Table 8: Evaluation of erasing two individual classes from Imagenette Shleifer & Prokop (2019).

| Method | Tench erased | | | | Parachute erased | | | |
|---|---|---|---|---|---|---|---|---|
| | $Acc_e \downarrow$ | $Acc_g \downarrow$ | $Acc_s \uparrow$ | $H_o \uparrow$ | $Acc_e \downarrow$ | $Acc_g \downarrow$ | $Acc_s \uparrow$ | $H_o \uparrow$ |
| SD-Neg. Ho & Salimans (2022) | 26.50 | 0.00 | 47.17 | 66.95 | 3.50 | 0.00 | 98.33 | 98.26 |
| FMN Zhang et al. (2024b) | 50.00 | 5.17 | 47.50 | 58.14 | 15.00 | 2.83 | 93.67 | 91.65 |
| ESD Gandikota et al. (2023) | 10.50 | 5.00 | 42.17 | 66.06 | 4.50 | 1.00 | 86.33 | 93.30 |
| SPM Lyu et al. (2024) | 54.00 | 2.33 | 48.67 | 57.12 | 11.50 | 2.83 | 99.33 | 94.76 |
| SalUn Fan et al. (2023) | 47.00 | 1.83 | 42.83 | 57.25 | 3.00 | 0.00 | 100.00 | **98.98** |
| AdvUnlearn Zhang et al. (2024c) | 0.50 | 1.83 | 44.00 | 69.83 | 1.00 | 3.50 | 72.17 | 87.41 |
| Ours | 30.00 | 2.83 | 79.33 | **80.68** | 9.50 | 1.50 | 97.83 | 95.47 |

art style removal, focusing on Van Gogh style. Each method was listed along the x-axis, with the y-axis showing mean ratings from 1 to 9. Ours attained the highest average of around 8.2, while most baseline methods Ho & Salimans (2022); Schramowski et al. (2023); Zhang et al. (2024b); Kumari et al. (2023); Gandikota et al. (2023); Lyu et al. (2024); Gandikota et al. (2024); Zhang et al. (2024c) scored below 6.

### A.2.2 OBJECT REMOVAL

**User study.** Table 8 presented the object removal performance for two target classes, "Tench" and "Parachute", evaluated using three metrics: efficacy ($Acc_e$), generality ($Acc_g$), and specificity ($Acc_s$). Detailed descriptions of the metrics and implementation details were provided in Section A.1.2. For the "Tench" class, our method achieved the highest overall score with $H_o = 80.68$, demonstrating a strong balance between target removal and object preservation. While some baseline such as AdvUnlearn Zhang et al. (2024c) achieved relatively low $Acc_e$ values, indicating strong erasure, it suffered from poor $Acc_s$, showing limited generalization performance. In contrast, our method maintained a high $Acc_s = 79.33$ while effectively suppressing both direct and synonymous mentions of the erased class. In the "Parachute" class, our method again demonstrated competitive performance with $H_o = 95.47$. These results demonstrated that our reference-based concept erasure method not only removed target objects effectively but also preserved unrelated content and generalized better than existing baselines.

### A.2.3 EXPLICIT CONTENT REMOVAL

**Quantitative evaluation.** Table 9 presented the comparative results of explicit content suppression across various baseline methods. Our method achieved the highest CLIP score (0.8231), indicating superior preservation of text-image alignment after concept removal. While SPM Lyu et al. (2024) achieved the lowest FID, our method ranked second (7.1039), demonstrating strong image quality alongside effective suppression. Notably, although SPM Lyu et al. (2024) showed better FID, its object removal efficacy was clearly lower than ours (Tabs.2, 8 and Fig. 7), confirming that our method offered a more favorable trade-off between concept removal effectiveness and image quality. These results highlighted the robustness and practicality of our embedding-guided approach without requiring model retraining. Table 4 presented a quantitative comparison of nudity removal performance, based on the number of exposed body part classes detected by NudeNet Praneeth (2019). The evaluation covered eight nudity categories—Armpits, Belly,

Table 9: Comparison of FID and CLIP scores for explicit content suppression.

| Method | FID↓ | CLIP↑ |
|---|---|---|
| SD | – | 0.8148 |
| Neg | 36.3273 | 0.8142 |
| ESD | 12.2535 | 0.7918 |
| FMN | 7.7915 | 0.8078 |
| SPM | **5.5389** | 0.8124 |
| UCE | 11.7670 | 0.8105 |
| Scissorhands | 148.6945 | 0.5952 |
| Salun | 41.8104 | 0.7390 |
| EraseDiff | 215.0143 | 0.4926 |
| AdvUnlearn | 32.6081 | 0.6693 |
| **Ours** | 7.1039 | **0.8231** |

Buttocks, Feet, Breasts (F), Genitalia (F), Breasts (M), and Genitalia (M)—and reported the total number of detections per method. A lower total count indicated a more effective suppression of explicit content. Detailed implementation details were provided in Section A.1.3. SD v1.4, without any content filtering, resulted in a high overall nudity detection count (3,293), serving as the baseline. Among text-based approaches, SLD Schramowski et al. (2023) and SPM Lyu et al. (2024) reduced the total detections to 2,176 and 1,919, respectively, but still left substantial explicit features. FMN Zhang et al. (2024b), despite being designed for mitigation, showed an even higher total count

Table 10: Idiosyncratic style removal vs. number of reference images.

| # of ref. images | FID↓ | CLIP↑ |
|---|---|---|
| 17 | 6.3087 | 0.8014 |
| 13 | 6.2230 | 0.8102 |
| 9 | 6.3418 | 0.8043 |
| **5** | **5.6319** | **0.8063** |
| 1 | 7.8826 | 0.7905 |

Table 11: Nudity removal vs. number of reference images.

| # of ref. images | FID↓ | CLIP↑ |
|---|---|---|
| 18 | 6.8892 | 0.8150 |
| 14 | 6.8911 | 0.8244 |
| **10** | **7.1039** | **0.8231** |
| 6 | 7.8662 | 0.7952 |
| 1 | 7.9005 | 0.7918 |

(3,656), suggesting poor generalization. ESD Gandikota et al. (2023) and UCE Gandikota et al. (2024) demonstrated improved suppression with totals of 776 and 739, respectively. Our proposed method achieved the best performance, with only 420 total detections, showing significantly lower counts in critical categories such as Genitalia (F) and Breasts (F). This highlighted the effectiveness of the proposed method in suppressing visually sensitive features and demonstrated that it outperformed all existing methods in terms of accuracy.

## A.3 Additional experimental results

### A.3.1 Effect of the reference Images

**Effect of the number of reference images.**
We investigated how the number of reference images affected the effectiveness of concept erasure by conducting ablation studies on two tasks with different conceptual scopes: idiosyncratic style removal and nudity removal. The results are presented in Tabs. 10 and 11. In the idiosyn-

Table 12: Comparison of robustness on different sets of reference images.

| $Acc_e$ ↓ | FID↓ | CLIP↑ |
|---|---|---|
| $3.9433 \pm 1.17$ | $8.0727 \pm 0.07$ | $0.8859 \pm 0.01$ |

cratic style removal task (10), we found that using more than five reference images did not lead to further improvements and occasionally degraded performance. The best results—lowest FID (5.6319) and highest CLIP score (0.8063)—were obtained with five reference images, suggesting that a small, focused set was sufficient to capture the core stylistic concept without overfitting. In contrast, the nudity removal task (11) benefited from a larger number of reference images. While the highest CLIP score (0.8244) was achieved with 14 images, the setting with ten reference images, which was used in the main paper (bolded), achieved a strong balance between FID (7.1039) and CLIP (0.8231). This indicated that broader concepts required more diverse visual representations for effective embedding. These results demonstrated that the optimal number of reference images varied by task. For narrow or visually consistent concepts, fewer examples sufficed, whereas broader or abstract concepts benefited from increased diversity. We thus adjusted the number of reference images per task to balance performance and practicality.

**Comparison of robustness on different sets of reference images.** We assessed the robustness of our method by evaluating its performance across different sets of reference images. Specifically, we conducted the object removal task multiple times using distinct reference image sets during the Concept Embedding Generation

Table 13: Comparison of robustness on different seeds.

| $Acc_e$ ↓ | FID↓ | CLIP↑ |
|---|---|---|
| $3.2767 \pm 1.67$ | $8.0710 \pm 0.08$ | $0.8816 \pm 0.02$ |

(CEG) stage. As summarized in Tab. 12, the results showed minimal variation across all evaluation metrics. These consistent results demonstrated that the proposed concept embedding remained effective regardless of the specific reference image subset used. For this experiment, we generated 200 images per setting to evaluate $Acc_e$, and 5,000 images per setting to compute FID and CLIP scores, ensuring sufficient statistical reliability. These findings indicated that our method maintained strong robustness under moderate variations in reference image composition, further supporting its applicability in practical scenarios where image selection may not be exact.

Table 14: Performance comparison of the transferability on van Gogh style removal.

| Model | Method | FID↓ | CLIP↑ | $S_{sty} \uparrow$ | $S_{obj} \uparrow$ |
|-------|--------|------|-------|------------|------------|
| SD v1.4 | Original | – | 0.8148 | $3.0369 \times 10^{12}$ | 0.2653 |
|  | Ours | 16.4156 | 0.7803 | $4.6519 \times 10^{16}$ | 0.2816 |
| SD v2.1 | Original | – | 0.8231 | $2.6645 \times 10^{11}$ | 0.2574 |
|  | Ours | 14.8826 | 0.8012 | $8.5788 \times 10^{15}$ | 0.2728 |
| SDXL | Original | – | 0.8305 | $6.5214 \times 10^{11}$ | 0.2791 |
|  | Ours | 12.4549 | 0.8097 | $1.8313 \times 10^{16}$ | 0.2814 |

Table 15: Overlapped vs. non-overlapped settings for idiosyncratic style removal.

| Method | FID↓ | CLIP↑ | $S_{sty} \uparrow$ | $S_{obj} \uparrow$ |
|--------|------|-------|------------|------------|
| **Ours (overlapped)** | 5.6319 | 0.8063 | $4.1083 \times 10^{16}$ | 0.2871 |
| **Ours (non-overlapped)** | 5.6714 | 0.8081 | $4.0872 \times 10^{16}$ | 0.2903 |

### A.3.2 Effect of random seeds

**Comparison of robustness on different seeds.** We further evaluated the robustness of our method by examining its sensitivity to random seed variation during generation. Using the same concept embedding, we repeated the object removal task with three different random seeds for the sampling process. As shown in Tab. 13, the performance across seeds remained consistent, with 5,000 images for evaluating FID and CLIP, ensuring statistical reliability across metrics. These results indicated that our embedding-guided inference framework was stable even under random noise variations introduced during sampling. The minimal fluctuation across metrics demonstrated that our approach did not rely on seed-specific artifacts, but instead robustly suppressed the target concept through consistent guidance. Combined with the findings from Tab. 12 (reference set variation), this confirmed the overall stability of our method under realistic perturbations.

### A.3.3 Transferability to other Diffusion models

To validate the applicability of our method across diverse generative architectures, we extended our evaluation to newer models, including Stable Diffusion v2.1 and SDXL, using van Gogh style removal as a representative task. As summarized in Tab. 14, our method consistently improved all performance metrics—FID, CLIP, style preservation score ($S_{sty}$), and object preservation score ($S_{obj}$)—compared to the original (unmodified) baselines across all tested backbones, indicating highly effective style suppression while maintaining overall generation quality. These results demonstrated that our embedding-driven erasure method was robust and transferable to more recent and complex diffusion architectures. Since our approach relied solely on the text encoder and did not require modification or retraining of the diffusion model itself, it was inherently compatible with a wide range of models. Moreover, as newer diffusion systems adopt more powerful vision–language encoders, the performance of our method is expected to improve further without any architectural changes or fine-tuning. We plan to include additional qualitative comparisons for SD v2.1 and SDXL in the revised version, and ongoing evaluations on SD3 and Flux are underway. These results confirm the scalability and forward-compatibility of our method for future generative models.

### A.3.4 Experimental setup for idiosyncratic style

To ensure a fair and unbiased evaluation, we revisited the experimental setup used in our idiosyncratic style removal task. In the initial configuration, we allowed overlap between the images used for DreamBooth Ruiz et al. (2023) fine-tuning and those used in Concept Embedding Generation (CEG). While this setup was chosen to guarantee full stylistic representation in the embedding, we agreed that such overlap could raise concerns regarding experimental fairness. To address this, we generated a new set of five reference images that did not overlap with the DreamBooth training set. These images were sampled from the same style distribution using StyleDrop Sohn et al. (2023), ensuring comparable visual characteristics without duplication. We then re-conducted the idiosyncratic style removal

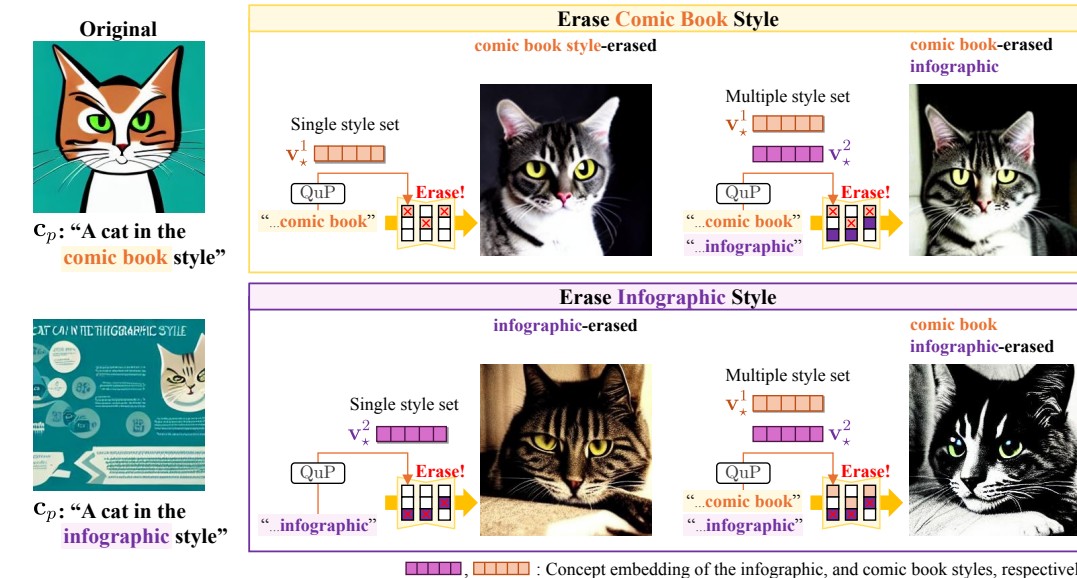

Figure 15: Qualitative results of multi-style removal. Our method successfully removed either the "Comic book style" or "Infographic style" from both single-style-erased and multi-style-erased models, demonstrating effective disentanglement and independent suppression of multiple styles.

experiments using the non-overlapped setting. As shown in Tab. 15, the results remained consistent across all metrics, including FID, CLIP, $S_{\text{sty}}$, and $S_{\text{obj}}$, with only minor differences observed. These findings confirmed that the effectiveness of our method was not reliant on overlapping samples and that the learned concept embedding retained robust performance even under stricter experimental conditions, supporting the fairness and generalizability of our setup.

### A.3.5 STYLE REMOVAL WITH MULTIPLE CONCEPT EMBEDDING

**Qualitative evaluation.** In this section, we provided further qualitative evaluations on the removal of multiple visual art styles, demonstrating the flexibility and modularity of our reference-based concept suppression method. Conventional text-based style removal methods handled the suppression of multiple concepts by combining multiple style descriptions—into a single negative prompt set

Table 16: Performance across different numbers of erased objects.

| # of erased object | $Acc_e \downarrow$ | FID↓ | CLIP↑ |
|---|---|---|---|
| 1 | 4.00 | 8.0850 | 0.8890 |
| 3 | 4.33 | 10.3328 | 0.8252 |
| 5 | 7.80 | 16.6773 | 0.7866 |
| 10 | 8.13 | 25.6975 | 0.7767 |

as ["style description 1, style description 2, . . ."]. This set was then used to steer the generation process away from producing the specified concepts. Inspired by this approach, we also aimed to suppress multiple concepts by independently training each concept embedding and aggregating them into a single model in the form of $[\mathbf{v}_\star^1, \mathbf{v}_\star^2, \ldots]$, thereby enabling unified multi-concept removal. Unlike previous sections that focused on comparisons with other style removal methods, this section highlighted the feasibility of removing multiple styles within a single model using our method. For any unmentioned implementation details, we followed the descriptions provided in the visual art style removal in Section A.1.1.

For the multi-style removal experiment, we designed a setup in which a single model could simultaneously remove both "Comic book style" and "Infographic style". We trained an individual concept embedding for each style using nine reference images per style. The two resulting embeddings were then injected into a single pre-trained SD model, enabling it to remove both styles within a unified framework. Figure 15 presented a qualitative evaluation and framework illustration demonstrating the feasibility of multi-style removal using our method. The leftmost column showed the original images generated from prompts "A cat in the comic book style" (top) and "A cat in the infographic style"

(bottom), each exhibiting strong stylization in line, color, and layout. The right-side panels visualized the results of style-specific removal from both single-style-erased models and a jointly trained multi-style-erased model. In the upper row, removing the comic book style from the corresponding single-style-erased model yielded a realistic cat image with stylization effectively suppressed. Importantly, the same degree of removal was observed when using a multi-style-erased model trained to erase both comic book and infographic styles simultaneously, indicating that the model retained the ability to selectively suppress the desired style (A), even in the presence of another erasable style (B). Similarly, the bottom row depicted the removal of the infographic style. The single-style-erased model successfully removed flat textures and diagrammatic cues. Notably, the multi-style-erased model also achieved comparable suppression when only the infographic style (B) was targeted, despite having learned to erase two styles (A and B). The erased-style indicators beside each image confirmed that the correct style(s) had been selectively deactivated at inference time. These results collectively validated that our method enabled the model to disentangle and suppress multiple styles independently, demonstrating flexible, composable multi-style removal within a unified framework.

Based on these results, our proposed method allowed individual concept embeddings to be trained for each removable concept and accumulated within a single model. These accumulated embeddings functioned as a concept embedding library, enabling the model to flexibly respond to different prompts. During inference, when a specific style or concept was mentioned in the input prompt, the corresponding concept embedding was automatically activated, guiding the model to suppress only the indicated style or concept. This allowed the model to internally store multiple embeddings while selectively applying only the relevant ones, depending on the prompt. Such a structure enabled flexible and scalable concept removal without requiring model retraining and demonstrated the potential to evolve into an extensible style removal framework. Consequently, this highlighted the feasibility of deploying our method in large-scale applications.

**Quantitative evaluation.** We additionally explored the simultaneous removal of multiple concepts by combining their embeddings. To systematically evaluate this setting, we conducted experiments on the Imagenette Shleifer & Prokop (2019) dataset, incrementally increasing the number of erased object classes in alphabetical order. As shown in Tab. 16, our method successfully erased multiple target objects without requiring retraining of the diffusion model, demonstrating its scalability beyond single-concept erasure. However, we observed a clear trade-off: as the number of erased concepts increased from one to ten, FID steadily rose and CLIP scores gradually decreased. These trends reflected the increased difficulty of simultaneously suppressing multiple visual concepts, especially when they are semantically diverse or visually overlapping. Despite this challenge, our results confirmed that multi-concept erasure was achievable using simple embedding fusion. To further improve performance and reduce degradation, we plan to explore adaptive weighting schemes and embedding disentanglement techniques in future work.

## A.4 RELATED WORK

**Image cloaking.** To prevent the unauthorized imitation of an artist's personalized style, adversarial examples (AE)—images modified with adversarial perturbations—have been proposed Shan et al. (2023); Liang et al. (2023); Xu et al. (2024); Chen et al. (2023); Poursaeed et al. (2018). This approach was developed for image distributors to use when publishing copyright-sensitive content online. AE are designed to interfere with downstream processes such as fine-tuning Hu et al. (2024); Ban & Dong (2022); Zhou et al. (2023); Liang et al. (2023) or image-to-image translation Van Le et al. (2023), thereby hindering the generation of outputs that reflect the original style Liang & Xiao (2023); Ge et al. (2023); Hönig et al. (2024); Song et al. (2018). Specifically, Liang & Wu (2023); Salman et al. (2023) block the generation of a copyrighted target style and instead induce the model to output a predefined anchor style. Zhu et al. (2024); Xiang et al. (2024); Wang et al. (2024) replace the original style with a watermark image designated by the content owner, or deliberately degrade the overall visual quality of the generated image Cui et al. (2025); Ahn et al. (2024). However, image cloaking techniques inevitably compromise visual quality due to the perturbations injection, and cannot be retroactively applied to images that have already been exposed online and used for training. Moreover, such approaches place the entire burden of copyright protection on the artist or image distributor, who must manually modify their works to prevent misuse. However, the responsibility to prevent copyright infringement should also lie with the developers and distributors of generative models.

Alongside technical and regulatory measures, there is a need for research into concept suppression techniques that can prevent copyrighted styles from being reproduced by generative models.

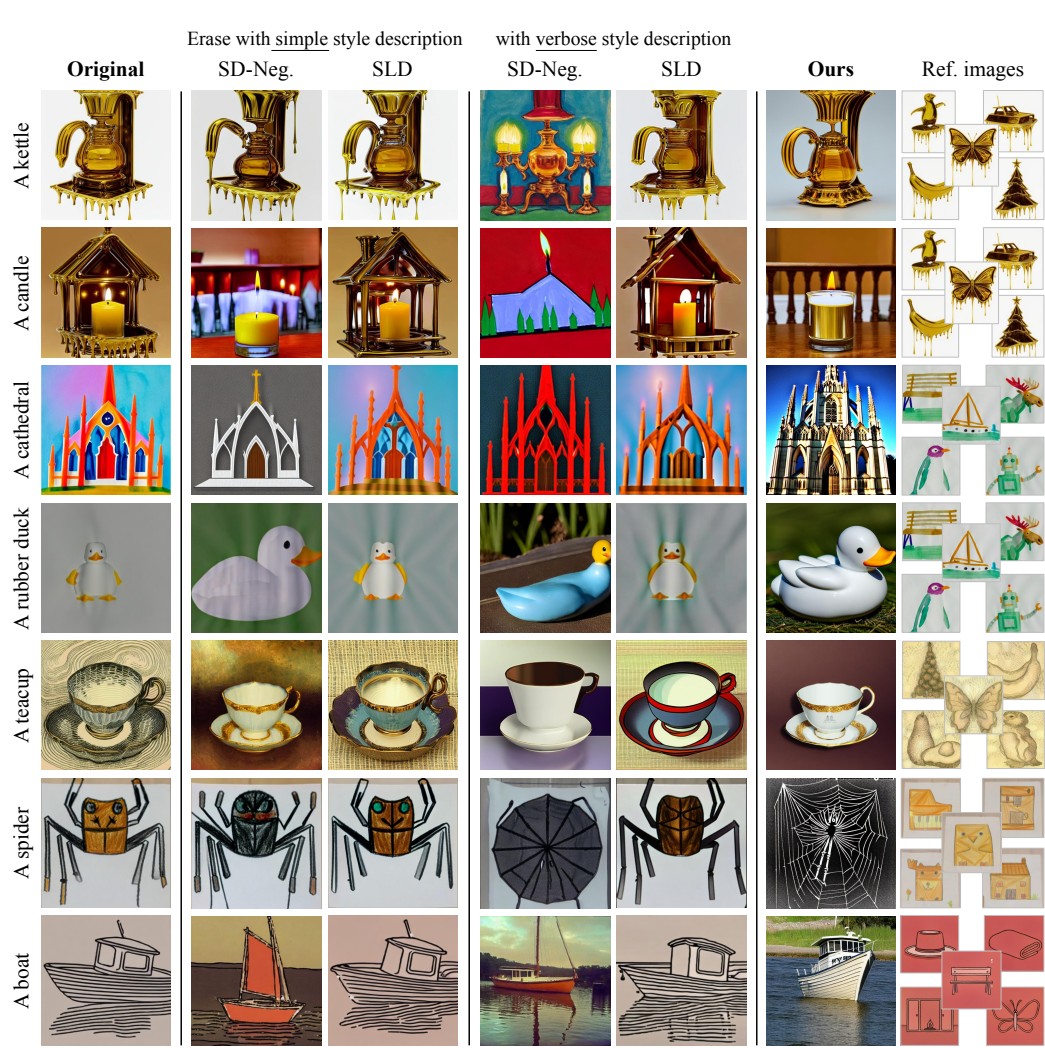

Figure 16: Qualitative comparison of idiosyncratic style removal across methods. Each row showed a different object category, while columns represented various style removal methods. The original images (leftmost) were generated using DreamBooth Ruiz et al. (2023) with a style token. Results from text-based methods (SD-Neg. Ho & Salimans (2022) and SLD Schramowski et al. (2023)) using simple or verbose style descriptions were shown next. Our method (sixth column) removed the target style using the concept embedding from the reference images (rightmost). The detailed style descriptions used were presented in the same order as the styles listed in Tab. 5.

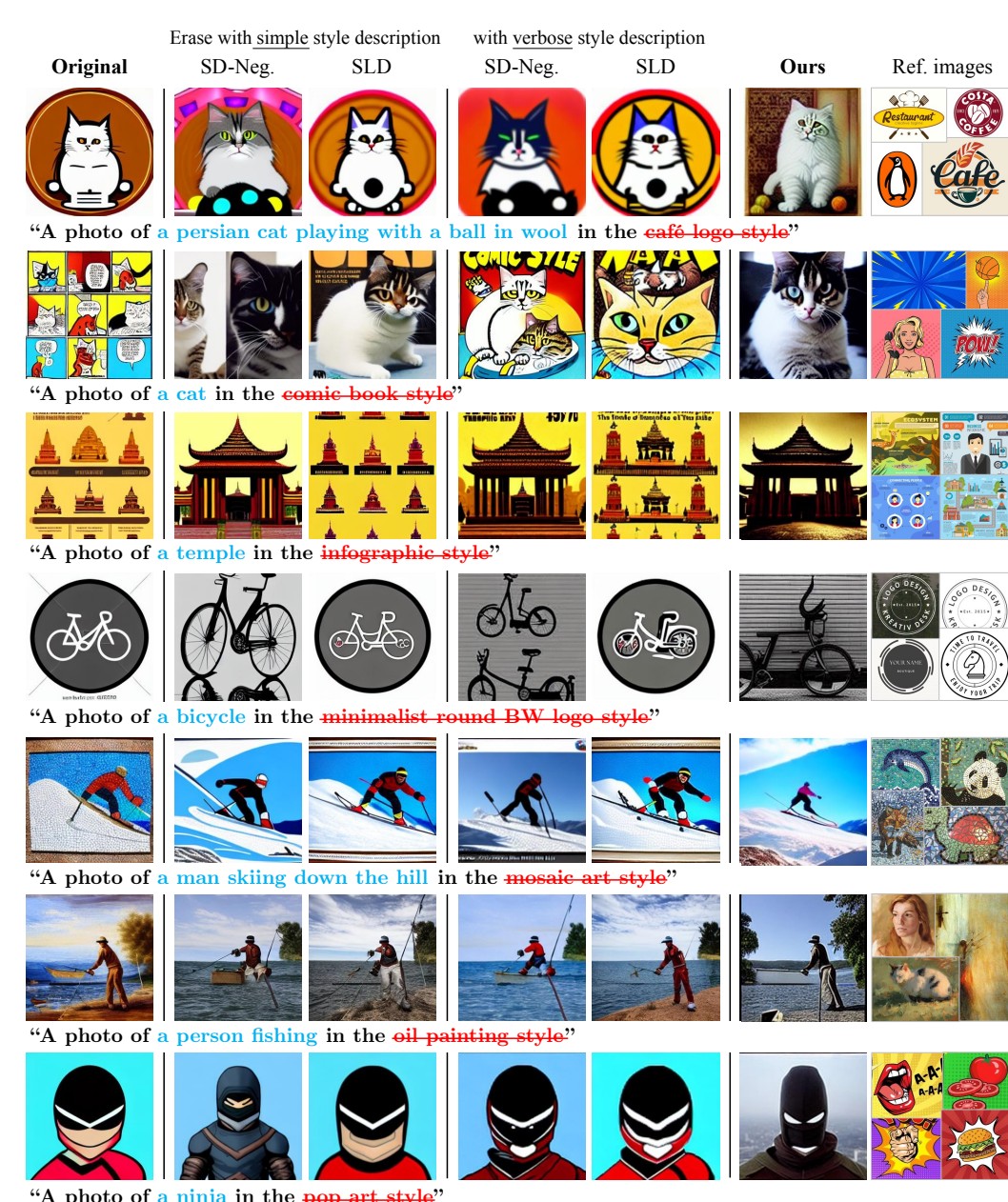

Figure 17: Qualitative comparison of visual art style removal across different methods. Each row corresponded to a different image prompt containing a specific object embedded within a distinct visual art style, such as "café logo style", "comic book style", or "oil painting style". The leftmost column showed the original images generated using SD v1.4. The next four columns displayed results from text-based style removal methods—SD-Neg. Ho & Salimans (2022) and SLD Schramowski et al. (2023)—applied with either simple or verbose style descriptions. The sixth column showed the results of our proposed method, which removed the target style using concept embeddings derived from the reference images shown in the rightmost column. The detailed style descriptions used were presented in Tab. 6.

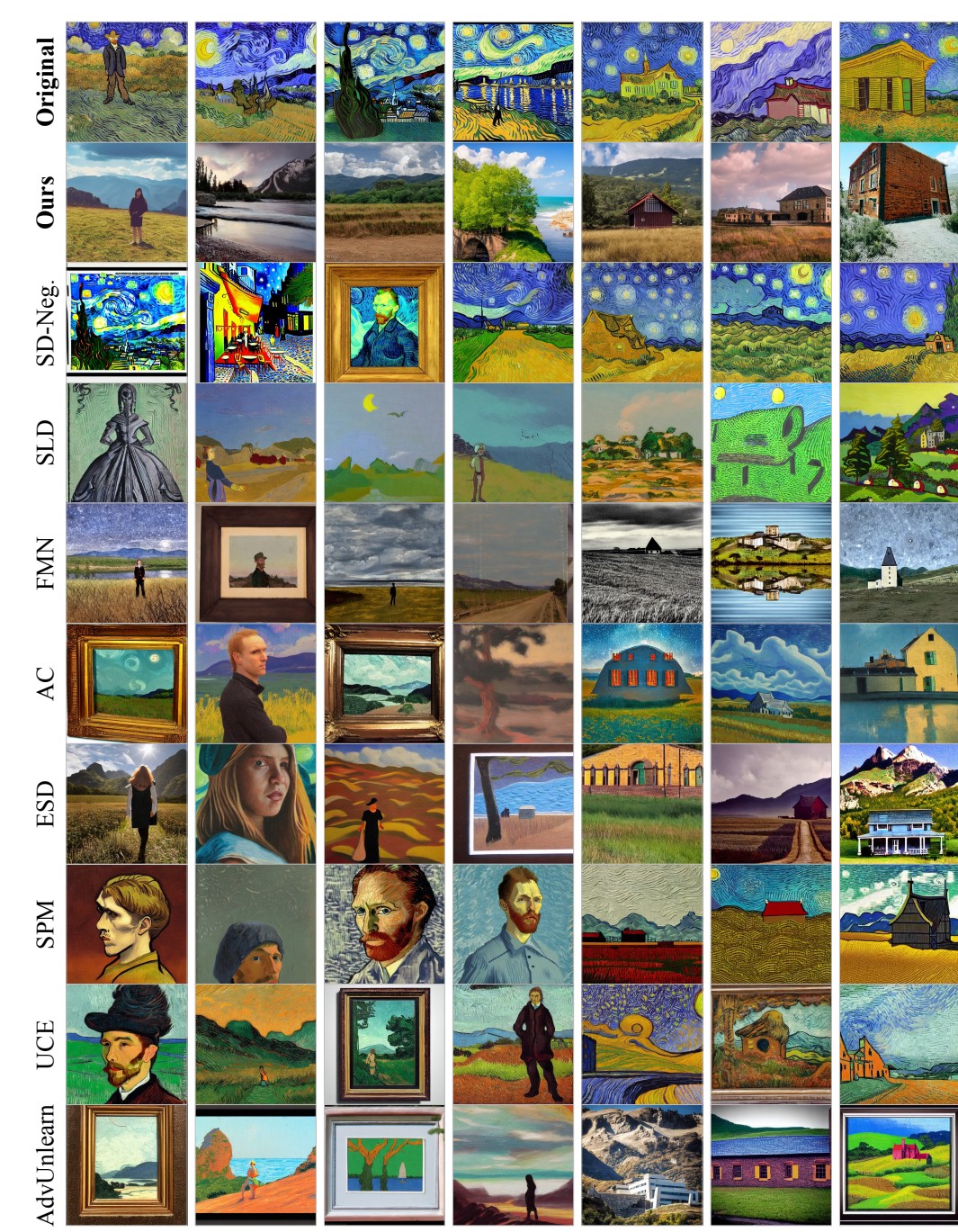

Figure 18: Qualitative comparison of Van Gogh style removal performance across various methods. The left four columns were generated using the prompt "A portrait in a scenic environment by Vincent van Gogh style," and the right three columns were generated using the prompt "A building in a stunning landscape by Vincent van Gogh style." The top row (Original) shows images generated directly from these prompts without any style removal. The subsequent rows display the outputs of different style removal methods Ho & Salimans (2022); Schramowski et al. (2023); Zhang et al. (2024b); Kumari et al. (2023); Gandikota et al. (2023); Lyu et al. (2024); Gandikota et al. (2024); Zhang et al. (2024c).

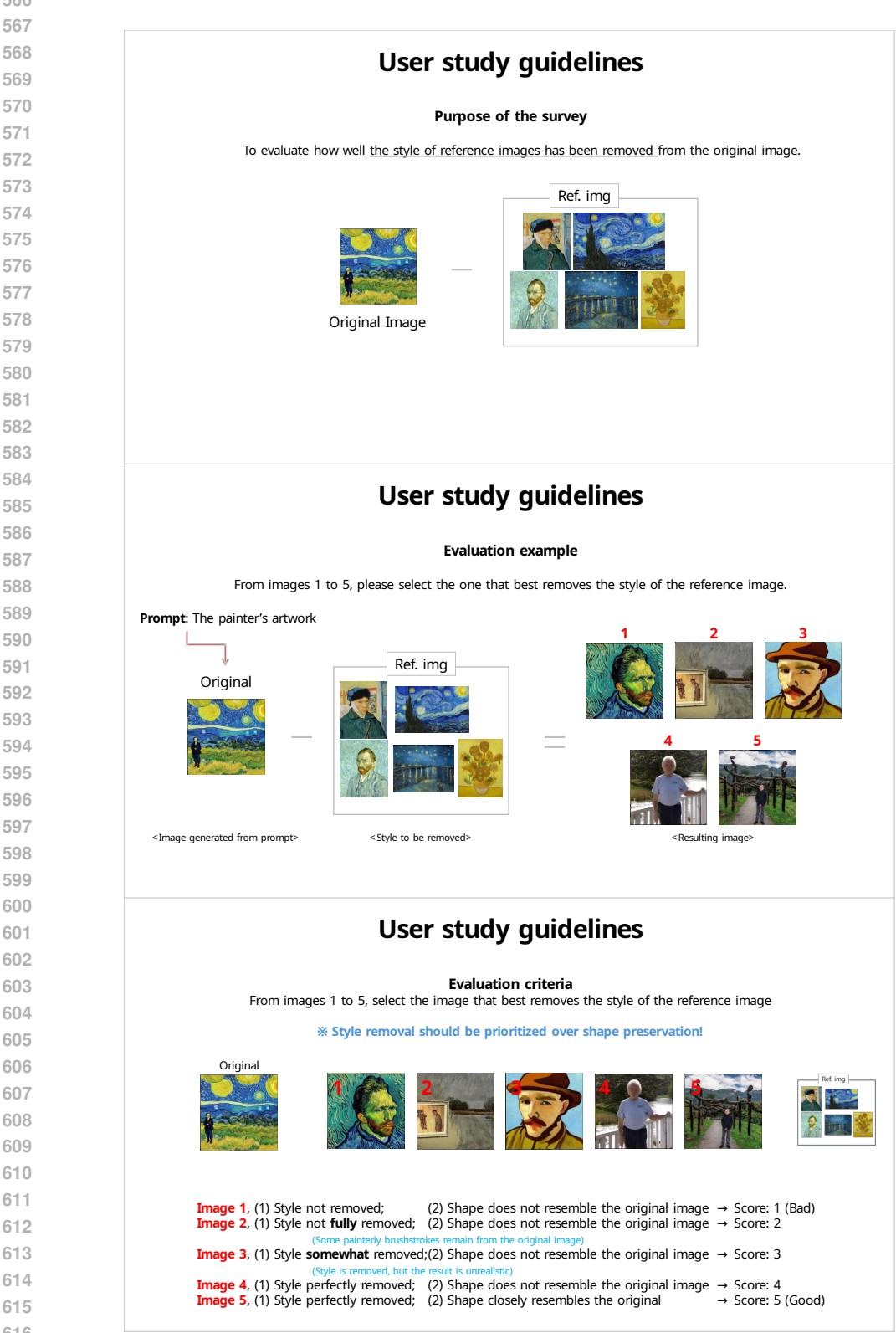

Figure 19: Guidelines provided to participants in the user study evaluating the performance of style removal methods.

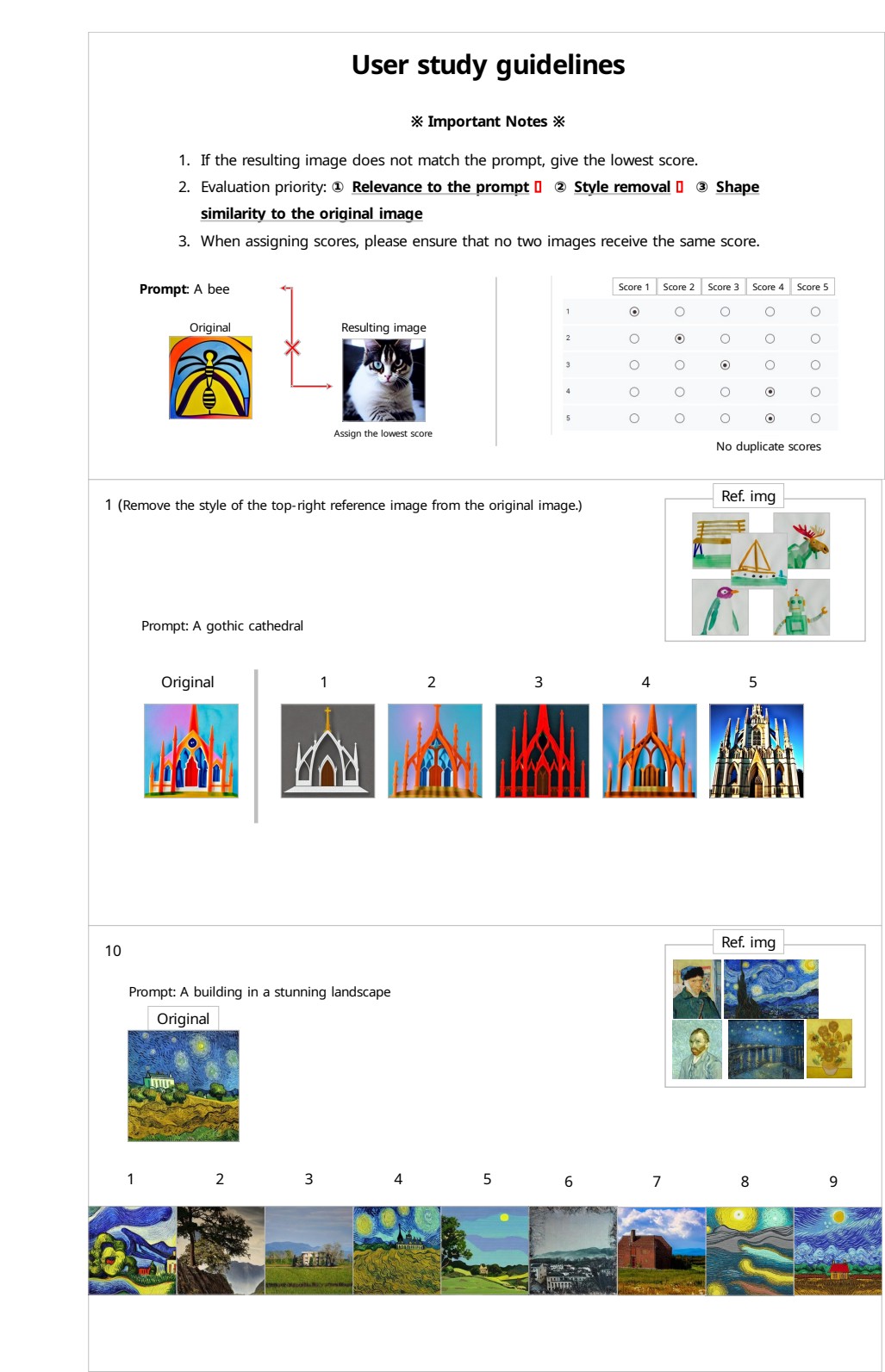

Figure 20: Guidelines (top row) and interface (rows 2-3) provided to participants in the user study evaluating style removal methods.

