# OpenReview forum: "The Quiet Prompt: Erasing Ineffable Styles from Diffusion via Concept Embedding"
_ICLR.cc/2026/Conference — Submitted to ICLR 2026_

### Official Review · Reviewer_NwdU · 2025-10-25

**Soundness:** 3
**Presentation:** 2
**Contribution:** 3
**Rating:** 4
**Confidence:** 3

**Summary:**

This paper proposes Quiet Prompt (QuP), a reference image-based concept erasure method designed to address the ineffectiveness of text-based negative prompts in removing complex and "ineffable" styles. The method learns a target concept embedding, $v^*$, by performing a process similar to Textual Inversion (termed Concept Embedding Generation, CEG) on a small set of reference images. Subsequently, during the inference stage, this embedding is incorporated as negative guidance (Concept-aware Negative Guidance, CNG) into a modified Classifier-Free Guidance (CFG) formulation. The authors demonstrate its qualitative advantages on various style and concept removal tasks and introduce a "style score" for evaluation.

**Strengths:**

1.  The method is novel. By using image references, it bypasses the limitations of text-based prompts, enabling the capture and suppression of complex style details that are difficult to describe accurately with language.
2.  The proposed method demonstrates superior performance in eliminating style residues and artifacts, with experimental results showing significant improvements over text-based approaches.
3.  The paper introduces two new metrics, $S_{sty}$ and $S_{obj}$, to measure the effectiveness of concept erasure.

**Weaknesses:**

1.  The paper's claim of being "training\-free" (line 102) is potentially misleading, as the process of generating the concept embedding vector ($v^\*$) from a few reference images is not entirely training\-independent.
2.  It is unclear if the model, once deployed, can only operate with pre\-learned $v^\*$ embeddings. In other words, to erase a new, previously unseen concept, is it necessary to collect corresponding reference images and retrain (or fine\-tune) a new $v^\*$ ?
3.  The method's robustness to **non\-target attributes** (e.g., watermarks, background textures, or irrelevant objects) within the reference images is a concern. If such details are inadvertently encoded into $v^\*$, how does the method ensure the embedding captures only the target concept and not these coincidental features? During inference, such a "contaminated" $v^\*$ might lead the model to erase both the target style and irrelevant background or objects, causing an over-erasure problem.
4.  The newly proposed metrics, $S_{sty}$ and $S_{obj}$, appear to be primarily based on empirical design. The paper lacks sufficient theoretical derivation or statistical justification to fully support their validity and interpretability.

**Questions:**

See Strengths and Weaknesses

---

### Official Review · Reviewer_iRPL · 2025-10-26

**Soundness:** 3
**Presentation:** 3
**Contribution:** 2
**Rating:** 2
**Confidence:** 3

**Summary:**

This paper proposes Quiet Prompt (QuP), a technique for concept erasure in diffusion models that avoids full retraining. QuP introduces two components: Concept Embedding Generation (CEG), which learns a latent embedding representing the concept to erase via Textual Inversion, and Concept-aware Negative Guidance (CNG), which uses this embedding as a negative semantic direction during diffusion sampling. The authors evaluate QuP on style, object, and NSFW content removal tasks and report improvements over existing unlearning baselines.

**Strengths:**

1. The method integrates seamlessly with existing diffusion pipelines.
2. Enables selective removal of copyrighted or unsafe styles.
3. Demonstrates quantitative and qualitative gains over SLD, ESD, and AdvUnlearn.

**Weaknesses:**

1. "Training-free" claim is misleading: The paper repeatedly emphasizes that the proposed approach is “training-free.” However, the Concept Embedding Generation (CEG) step clearly involves gradient-based optimization to learn new embedding vectors, which constitutes a form of training—even if limited in scope. This characterization risks misleading readers. Furthermore, the paper omits any discussion of computational cost, convergence behavior, or sensitivity to hyperparameters (e.g., learning rate, number of iterations).

2. Lack of theoretical grounding: The method assumes that concept embeddings learned through textual inversion reside in the same semantic space as pre-trained text embeddings, but this assumption is not substantiated. There is no theoretical justification or empirical validation (e.g., t-SNE visualization, cosine similarity analysis, or linear probing) demonstrating that these learned embeddings meaningfully align with the textual semantic manifold.

3. Limited generalization across diffusion models. All experiments are conducted exclusively on Stable Diffusion v1.4, which limits the generality of the claims. There is no evidence that the method transfers to more recent or structurally different text-to-image models (e.g., SDXL, DeepFloyd, or non-CLIP-based backbones).

4. Poor robustness and scalability in multi-concept and adversarial scenarios. The method shows substantial degradation in multi-concept removal tasks (Table 16), suggesting limited scalability beyond single-concept editing. In addition, robustness to prompt evasion and adversarial reformulations (e.g., synonym substitution, blended or indirect prompts) is not evaluated. This omission raises concerns about the reliability of the proposed approach in real-world or safety-critical contexts.

**Questions:**

1. How consistent are CEG embeddings across seeds or reference sets?
2. Is negative guidance applied dynamically per-step or statically precomputed?

---

### Official Review · Reviewer_rnM8 · 2025-10-29

**Soundness:** 2
**Presentation:** 3
**Contribution:** 2
**Rating:** 4
**Confidence:** 4

**Summary:**

This paper raises an important question: how to erase personalized or purely visual concepts that lack clear textual descriptions. To tackle this challenge, the authors propose a new erasure framework capable of handling visually grounded and implicit concepts. Their method enables targeted removal of visual attributes while maintaining the model’s generative quality.
The proposed method consists of two stages: (1) Concept embedding generation, which compresses the color, texture, and morphology of an unwanted style into a single latent vector; and (2) Concept-aware negative guidance, which uses the extracted concept embedding as a semantic anchor to steer the diffusion process away from the undesirable concept without requiring model retraining.
Experiments demonstrate that the proposed approach achieves more precise and controllable erasure compared to existing text-based methods. The work provides a practical and scalable solution to improve the safety of text-to-image diffusion models.

**Strengths:**

1. This paper is well-structured and easy to follow.

2. The proposed embedding-based concept removal approach appears novel.

3. Extensive experimental results demonstrate that the method achieves superior performance.

**Weaknesses:**

1. First, I believe the proposed method does not clearly demonstrate its necessity in the context of style removal. As shown in Figures 3, 4, 6, and 7, style removal often leads to a loss of contextual fidelity. If removing the style also alters the image content, then the proposed method becomes less meaningful, since one could simply use a text prompt to regenerate a style-free image instead.

2. The paper does not compare or discuss existing works on style–content disentanglement, which are directly relevant to this task.

    [1] Uncovering the disentanglement capability in text-to-image diffusion models.

    [2] Less is More: Masking Elements in Image Condition Features Avoids Content Leakages in Style Transfer Diffusion Models.

    [3] Stylediffusion: Controllable disentangled style transfer via diffusion models.

    [4] Not only generative art: Stable diffusion for content-style disentanglement in art analysis.


3. The proposed approach relies on additional style reference images. It remains unclear how the method can effectively perform style removal when no explicit style reference is available.

4. In Line 102, the authors claim the method is training-free, but it is not explained how $v_{\star}$ is obtained.

5. Can the proposed method simultaneously perform both style removal and object removal?

**Questions:**

See weaknesses.

---

### Official Review · Reviewer_8PGp · 2025-10-29

**Soundness:** 2
**Presentation:** 2
**Contribution:** 2
**Rating:** 4
**Confidence:** 4

**Summary:**

This paper proposes a method for removing specific styles or concepts from pre-trained diffusion models. The key feature of the proposed method is that it leverages multiple images containing the target concept, rather than relying on text. The method consists of two main components: (1) Concept Embedding Generation, which embeds the target concept as soft tokens derived from multiple images, and (2) Concept-aware Negative Guidance, which prevents the generation of the target concept by subtracting the output associated with the soft tokens. Experimental results demonstrate that, compared with existing concept removal methods, the proposed method more effectively removes the target concept while better preserving other information.

**Strengths:**

1. The authors propose a style removal method that utilizes multiple images containing the target concept rather than text. While text-based removal methods such as Negative Prompting and Safe Latent Diffusion struggle with precise style specification, the proposed method enables detailed style removal by employing images.

2. Another advantage of concept removal using images is robustness against rephrasing. Concept removal using text may generate the intended image if synonyms of the target text are input. In contrast, the proposed method uses images, allowing it to avoid generating the image even when synonyms are input.

3. Experiments demonstrate that the proposed method removes style and concept more efficiently than existing methods. Style and concept removal techniques are crucial technologies for copyright protection and avoiding inappropriate content, indicating significant impact.

**Weaknesses:**

1. The difference between the proposed "Concept embedding generation" and Textual Inversion (Gal et al., 2022) is unclear. The Textual Inversion paper demonstrates the ability to convert concepts and styles from multiple images into token embeddings. The authors should clarify this distinction.

2. The difference between the proposed "Concept-aware negative guidance" and the methods "Negative Prompt" and "Safe Latent Diffusion" (Schramowski et al., 2023) is unclear. Both Negative Prompt and Safe Latent Diffusion, like the proposed method, aim to prevent the generation of specific concepts by negatively applying classifier-free guidance. The authors should clarify how their approach differs from these existing methods.

3. Whether removing a specific style influences other styles has not yet been verified. For instance, [1] investigates the effect of removing the "Van Gogh" style on the "Picasso" and "Monet" styles, and reports that Negative Prompt and Safe Latent Diffusion negatively impact the "Picasso" and "Monet" styles. Similar experiments should be performed to ensure that the proposed method does not unintentionally degrade other styles.

[1] Wang, Yuan, et al. "Precise, fast, and low-cost concept erasure in value space: Orthogonal complement matters." 2025 IEEE/CVF Conference on Computer Vision and Pattern Recognition (CVPR). IEEE, 2025.

**Questions:**

Please add explanations regarding the weaknesses.

---

### Official Review · Reviewer_yDAR · 2025-10-31

**Soundness:** 4
**Presentation:** 4
**Contribution:** 3
**Rating:** 6
**Confidence:** 3

**Summary:**

This paper proposes Quiet Prompt (QuP), a reference-based concept-erasure method for text-to-image diffusion. From fewer than ~10 reference images, it learns a compact concept embedding via textual inversion and then uses a concept-aware negative guidance during sampling to suppress that concept without retraining the base model. QuP targets “ineffable” styles (hard to describe in text) as well as objects and NSFW categories, and evaluates across style/object/NSFW erasure with quantitative metrics and user studies.

**Strengths:**

- Clear, practical mechanism (few-shot embedding + negative guidance) that’s easy to plug into existing T2I pipelines; strong qualitative illustrations.
- Broad evaluations: idiosyncratic & art styles, objects, and NSFW; combines quantitative metrics (FID/CLIP, counts) and user preference.
- Competitive numbers vs text-based baselines (e.g., SD-Neg/SLD) and useful ablations on data efficiency, robustness to reference sets/seeds, and cross-backbone transfer to SD2.1/SDXL.

**Weaknesses:**

- Novelty positioning needs sharper contrast. The core recipe—learn a concept embedding with textual inversion, then use it as a negative condition—sits close to prior personalization and prompt-based erasure. Please provide a crisper head-to-head against text-only negatives and image-guided negatives to establish distinctiveness.
  - [1] https://arxiv.org/abs/2211.12572
  - It represents the strongest “image-guided” alternative to your reference-learned negative. If QuP can erase the style/object as well as or better than PnP while preserving unrelated content (and with lower inference overhead), that cleanly establishes distinctiveness.
- Style-score validation. The proposed style metric is sensible but would benefit from correlation to human preference and a comparison to alternative style-distance measures (e.g., Gram/feature stats, Style-Aligned).
- Scope beyond SD1.4. Cross-model results (SD2.1/SDXL) live mostly in the appendix; consider elevating at least one row to the main tables to support generality claims.

**Questions:**

- The claims regarding the comment
  - It represents the strongest “image-guided” alternative to your reference-learned negative. If QuP can erase the style/object as well as or better than PnP while preserving unrelated content (and with lower inference overhead), that cleanly establishes distinctiveness.
- Why does a reference-learned concept embedding outperform (i) text-only negative prompts and (ii) image-guided negatives (e.g., CLIP-image embedding as negative)? Please add direct controls on the same splits as Table 1.
- Can you report correlation between your style distance and user preference (Fig. 14), and compare the distance to Gram/feature-statistic or Style-Aligned alternatives?
- For object erasure, can you show attention/attribution maps indicating suppression stems from the negative concept embedding rather than prompt drift (ties to Tables 2 & 8)?

---

### Official Review · Reviewer_xvUR · 2025-11-02

**Soundness:** 3
**Presentation:** 3
**Contribution:** 2
**Rating:** 2
**Confidence:** 4

**Summary:**

The paper proposes a method for removing hard-to-describe or abstract concepts from text-to-image diffusion models using a few reference images. It first learns a textual embedding through textual inversion from these images, then applies negative guidance with the learned embedding to suppress the target concept during generation. The approach also generalizes to standard objects, styles, and NSFW content. For evaluation, the paper fine-tunes a DreamBooth LoRA on the same reference images to learn the target style or concept, and then tests the proposed removal method on it

**Strengths:**

1. Being able to remove abstract concepts is indeed useful and practical. It’s often hard to describe concepts just via text and using techniques like textual inversion to first learn a concept embedding is both intuitive and shown to be effective by the paper.
2. The paper is well written and easy to understand

**Weaknesses:**

1. Negative guidance for concept removal has been explored (e.g., SD-Neg, SLD, Dynamic Negative Guidance). The main addition here is combining it with textual inversion. It would be great if the paper explores some unique advantages that using reference images provides—e.g., finer disentanglement of attributes (stroke style vs. color palette, content vs. texture). Demonstrating user-mentioned partial removals would make the contribution stronger.
2. The qualitative and quantitative evaluation can be improved. Figure 4 only shows the qualitative comparison with SLD and SD-Neg and not other fine-tuning-based methods. Also, adding comparison to more recent baselines, such as EraseAnything or MACE, in Table 2 would be great. Prior works (e.g., CA [Reverse KL divergence, Sec. 4.3]) also demonstrate concept removal using real reference images; adding a comparison to that would clarify the value of using real reference images vs. negative guidance for concept removal.
3. Does the method still work for guidance distilled models like FLUX.1-dev or few-step distilled models?

Minor points:
1. The paper first uses DreamBooth LoRA to train a model on abstract concepts and then uses their method to remove it. However, it would be more convincing to show the efficacy of the method on abstract styles that can be generated by the model itself.
2. It would be great to discuss the choice to measure KL divergence (Eq. 5) in VAE feature space rather than CLIP/DINO feature space.
3. Does the choice of reference images have any effect on the final performance, e.g., using a different set of reference images for DreamBooth LoRA training and the textual inversion part, and a different set of prompts for NSFW removal than I2P prompts?

[1] Dynamic Negative Guidance of Diffusion Models [https://openreview.net/pdf?id=vvBAZJh2nQ]
[2] EraseAnything: Enabling Concept Erasure in Rectified Flow Transformers

**Questions:**

Please look at the weakness section, particularly regarding any unique advantage that the use of textual inversion with real reference images provides in the proposed method

---

### Meta-Review · Area_Chair_Ssct · 2026-01-05

**Summary:**

The reviewers generally agree that the paper addresses an important and practical problem: erasing "ineffable" or abstract visual concepts from diffusion models that are difficult to describe via text. While the proposed mechanism—combining textual inversion with negative guidance—is praised for its simplicity and effectiveness , significant concerns remain regarding the paper's novelty and technical claims. Key issues include the clarity of distinction from existing methods such as Textual Inversion and Safe Latent Diffusion, the absence of several relevant recent baselines, a potentially misleading claim of the approach as “training-free”, robustness issues, and limited evidence of generalizability beyond Stable Diffusion v1.4. These suggest that the work may benefit from further development before being competitive for acceptance at ICLR.

**Reviewer Concerns:**

The authors did not submit the rebuttal.

**Reviewer Scores:**

As no rebuttal were submitted, the reviewers are likely to keep their original scores.

---

### Decision · Program_Chairs · 2026-01-26

Reject